# Efficient Exploration via Fragmentation and Recall

## Abstract

Efficient exploration and model-building are critical for learning in large state-spaces. However, agents can face problems like getting stuck in local optima during exploration, and catastrophic forgetting when constructing models in heterogenous environments. Here, we propose and apply the concept of *Fragmentation-and-Recall* to solve spatial (*FarMap*) and reinforcement learning problems (*FarCuriosity*). Agents construct local maps or local models, respectively, which are used to predict the current observation. High surprisal points lead to a fragmentation event. At fracture points, we store the current map or model fragment in a long-term memory (LTM) and initialize a new fragment. On the other hand, Fragments are recalled (and thus reused) from LTM if the observations of their fracture points match the agent's current observation during exploration. The set of fracture points defines a set of intrinsic potential subgoals. Agents choose their next subgoal from the set of near and far potential subgoals in the current fragment or LTM respectively. Thus, local maps and model fragments guide exploration locally and avoid catastrophic forgetting when learning in heterogeneous environments, while LTM promotes global exploration. We evaluate *FarMap* and *FarCuriosity* on complex procedurally-generated spatial environments and on reinforcement learning benchmarks to demonstrate that the proposed methods are more efficient from a memory usage standpoint, and achieve better task performance overall.

## 1 Introduction

Human episodic memory breaks our continuous experience of the world into episodes or fragments that are divided by event boundaries that involve large changes of place, context, affordances, and perceptual inputs (Baldassano et al., 2017; Ezzyat & Davachi, 2011; Newtson & Engquist, 1976; Richmond & Zacks, 2017; Swallow et al., 2009; Zacks & Swallow, 2007). The episodic nature of memory is a core component of how we construct models of the world. It has been conjectured that episodic memory makes it easier to perform memory search, and to use the retrieved memories in chunks that are relevant for the current context. Humans also continue to learn and memorize new information throughout their lives, without needing to reconfigure all previously stored memories. These observations suggest a certain locality or fragmented nature to how we model the world.

Chunking of experience has been shown to play a key role in perception, learning and cognition in humans and animals (De Groot, 1946; Egan & Schwartz, 1979; Gobet et al., 2001; Gobet & Simon, 1998; Simon, 1974). In the hippocampus, place cells appear to chunk spatial information by defining separate maps when there has been a sufficiently large change in context or in other non-spatial or spatial variables, through a process called *remapping*; see Colgin et al. (2008); Fyhn et al. (2007). Grid and place cells in the hippocampal formation have also been shown to fragment their representations when the external world or their own behaviors have changed only gradually rather than discontinuously (Derdikman et al., 2009; Low et al., 2021). Recently, Klukas et al. (2021) proposed how such remapping could occur in even during continuous navigation through a continuous environment, modeling the process as one of online clustering based on observational surprisal.

Similarly, when fitting complicated manifolds or functions, it is common to build a set of simpler local models of the manifold or function. Inspired by these ideas, here we propose building models for complicated spaces by fitting a sequence of local models, and using local models obtained through an online process of fragmentation to aid in the the exploratory process of moving through a large space and building a model of the space.

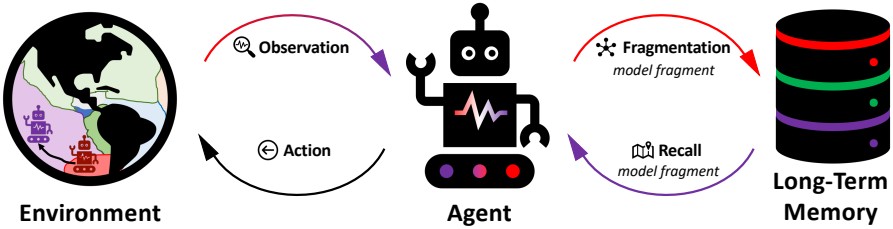

Figure 1: Overview of our approach. Given an observation from the environment, the *FarMap* or *FarCuriosity* agent decides whether to fragment the space based on how well it can predict the observation. If *fragmentation* occurs, the current map (or model) fragment is stored in long-term memory (LTM); the agent then initializes a new map (or model) fragment. Conversely, if the current observation closely matches with the observations stored in LTM, the agent loads an existing map (or model) fragment from there (*recall*). Based on the current fragment, the agent selects an action to explore the environment.

We propose a new framework for exploration based on a concept of online *Fragmentation-and-Recall*, schematized in Figure 1. This model combines two ideas: 1) when faced with a complex world, it can be more efficient to build and combine multiple (and implicitly simpler) local models than to build a single global (and implicitly complex) model, and 2) boundaries between local models should occur when a local model ceases to be predictive.

In what follows, as an agent explores, it predicts its next observation. Based on a measure of surprisal between its observation and prediction, there can be a *fragmentation* event, at which point the agent writes the current model into long-term memory (LTM) and initiates a new local model. While exploring the space, the agent consults its LTM, and *recalls* an existing model if it matches its observations. For the spatial domain, this is very similar to Klukas et al. (2021). The agent uses its current local model to act locally, and its LTM to act more globally. We apply this concept to solve spatial exploration and more general reinforcement learning exploration problems, and call the corresponding approaches FarMap and FarCuriosity, respectively.

We evaluate the proposed framework on procedurally-generated spatial environments and reinforcement learning benchmarks. Experimental results support the effectiveness of the proposed framework; FarMap explores the spatial environment with much less memory usage and is faster than its baselines (Yamauchi, 1997) by large margins, and FarCuriosity achieves better performance than the baseline fragmentation-less curiosity module (Burda et al., 2019) on standard heterogeneous Atari game benchmarks[1]. The contribution of this paper is three-fold as follows:

- We propose a new framework for exploration based on *Fragmentation-and-Recall* that divides the exploration space into multiple fragments and recalls previously explored ones.
- We implemented our framework in spatial exploration tasks, referring to it as FarMap with short and long-term memory. Our experiments showed that FarMap reduces online memory size and wall-clock time relative to baselines.
- We implemented our framework in a curiosity-driven reinforcement learning exploration setting, referring to it as FarCuriosity. FarCuriosity avoids catastrophic forgetting and achieves better performance compared to the baseline in heterogenous environments.

## 2 RELATED WORK

**Frontier-based spatial exploration in SLAM**   SLAM (simultaneous localization and mapping) agents must efficiently explore spaces to build maps. A standard approach to exploration in SLAM is to define the *frontier* between observed and unobserved regions of a 2d environment, and then select exploratory goal locations from the set of frontier states (Yamauchi, 1997). Frontier-based exploration has been extended to 3d environments (Dai et al., 2020; Dornhege & Kleiner, 2011) and used as a building block of more sophisticated exploration strategies (Stachniss et al., 2004). Although conceptually simple, frontier-based exploration can be quite effective compared to more sophisticated decision-theoretic exploration (Holz et al., 2010). A cost of frontier-based exploration is

---

[1]Heterogeneous environment is an environment that has diverse states that require the larger model capacity to memorize visited states for generating intrinsic reward. Please refer to Section 4.2 for more details.

the use of global maps and global frontiers, which makes the process memory-expensive and search intensive. In contrast to frontier-based exploration, our approach *learns* the surprising parts of an environment as intrinsic subgoals, selecting among those as the exploratory goals.

**Submap-Based SLAM**   Submap-Based SLAM algorithms involve mapping a space by breaking it into local submaps that are connected to one another via a topological graph. Such Submap-Based SLAM methods are usually designed to avoid the problems of accruing path integration errors when building maps of large spaces (*e.g.* SegSLAM) (Fairfield et al., 2010) and to reduce the computational cost of planning paths between a start and target position (Maffei et al., 2013; Fairfield et al., 2010; Klukas et al., 2021). Segmented DP-SLAM (Maffei et al., 2013) adds DP-SLAM (Eliazar & Parr, 2003) to SegSLAM for reducing the search space, generating segments periodically at fixed time-intervals. Topological SLAM (Choset & Nagatani, 2001) generates new landmarks in an environment to build a topological graph of the landmarks, and navigates based on the graph. Our Fragmentation and Recall method, in particular FarMap, is closely related to these methods in that we build multiple submaps. However, FarMap divides space based on properties of the space (how predictable the space is based on the local map or model), and does so in an online manner using surprisal.

**Curiosity-Driven Reinforcement Learning**   Motivated by the insufficiency of rewards alone to guide exploration in sparsely rewarded RL environments, several works construct *intrinsic rewards* encouraging agents to explore unfamiliar parts of an environment. Count-based methods provide intrinsic rewards for reaching infrequently visited states. These methods have strong theoretical guarantees for multi-arm bandits (Auer, 2003), and can be effective for reinforcement learning (Bellemare et al., 2016; Tang et al., 2017; Ecoffet et al., 2021). Prediction-based methods estimate where the agent cannot predict some aspect of the environment, and provide intrinsic rewards for visiting those states (Burda et al., 2019; Choshen et al., 2018; Houthooft et al., 2017; Pathak et al., 2017; Raileanu & Rocktäschel, 2020). Pathak et al. (2017), for instance, learns a forward model to predict the next state given the current state-action pair, and provides rewards scaling with the state's prediction error.

**Memory-Based Reinforcement Learning**   Memory-based reinforcement learning aims to solve the long-term credit assignment problem. Hung et al. (2019) combine LSTM (Hochreiter & Schmidhuber, 1997) with external memory along with an encoder and decoder for the memory. Lampinen et al. (2021) use a hierarchical LTM with chunks and attention for long-term recall inspired by Transformers (Vaswani et al., 2017); however, their chunks are formed periodically rather than based on content and are not used as intrinsic options for exploration. Memory is also used for improving exploration: Go Explore (Ecoffet et al., 2021) remembers promising states so that it can directly visit those states, where the definition of promising can include rewarded states and frontiers. Similar to our proposal, we may view promising states in GoExplore as intrinsic options; however, if the promising states are frontiers, GoExplore becomes similar to frontier-based SLAM methods without fragmentation. Savinov et al. (2019) memorizes visited states for calculating reachability to penalize to an action that leads to reachable future states. While earlier memory-based exploration methods use memory as a guide for evaluating the intrinsic reward of a state, we augment existing exploration methods with memory, in combination with map or model fragmentation, to more efficiently explore.

## 3   FRAGMENTATION AND RECALL FRAMEWORK

The proposed framework is based on the notion of fragmentation and recall: While exploring an environment, an agent builds a local model and uses the local model in short-term memory (STM) to compute a surprisal signal that depends on the current observation and the agent's local model-based prediction. The surprisal can be any uncertainty estimate such as negative confidence or future prediction error. When the surprisal exceeds some threshold, this corresponds to a *fragmentation* event. At a fragmentation event, the local model is written to long-term-memory (LTM) building a connectivity graph that relates model fragments to each other so that it can provide information across the current local model without direct access to the stored models in LTM. The current (abstracted) observation (*fracture point*) is also stored, and the agent initializes an entirely new local model. On the other hand, if the current observation is sufficiently similar to a stored fracture point in LTM, the agent *recalls* the corresponding model fragment (local model). Hence, the agent can preserve and reuse previously acquired information. Sections 3.1 and 3.2 describe two application domains; spatial exploration and curiosity-driven reinforcement learning, respectively.

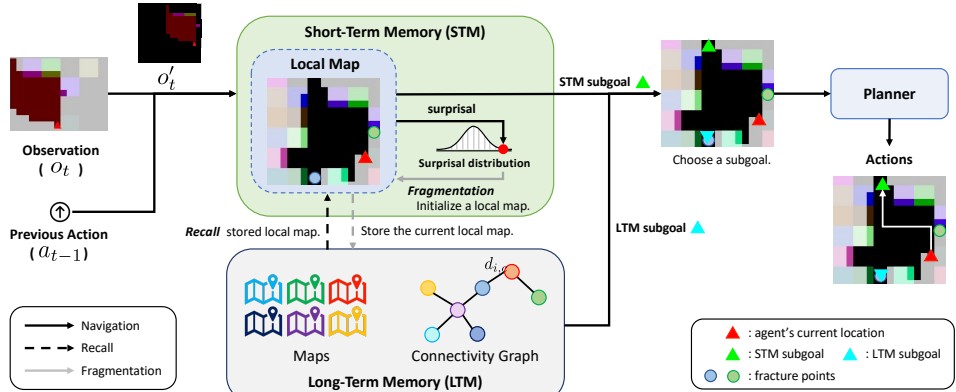

Figure 2: Illustration of the FarMap framework. Navigation (black arrow): Given the current observation which is ego-centric top-down view with restricted field of view and previous action, the agent updates its short-term memory (STM) and selects a subgoal from the current local map in STM or the local map connectivity graph stored in LTM. The planner generates a sequence of the actions for the shortest-path to the subgoal. Recall (dashed arrow): If the agent arrives at a fracture point (circle in the map), a corresponding local map is recalled from LTM and the current local map stored in LTM is updated. Fragmentation (gray arrow): If the current surprisal is higher than a threshold, the current local map is stored in LTM and a new local map is initialized. $o'_t$ is spatially transformed observation with the same size of the current local map to update the map.

## 3.1 FRAGMENTATION AND RECALL IN SPATIAL EXPLORATION (FARMAP)

SLAM algorithms explore space efficiently by aiming to explore frontier regions. They categorize each region (cell) as known and unknown based on whether it is previously observed or not and occupied and unoccupied (empty) based on its occupancy. Unknown cell adjacent to empty cells is called frontier. Typically frontier-based exploration involves frequent consultation of a global map. The memory and search cost of finding subgoals grows rapidly with environment size; for agents exploring a very large space, the computational costs could explode. Here, we propose a fragmentation-and-recall based exploration strategy for spatial exploration (*FarMap*). In *FarMap*, each local model is a spatially local map built in short-term memory (STM). As Figure 2 shows, given the observation and the previous action, the FarMap agent updates the current local map. The agent decides its next subgoal, selecting a point within the current local map or a fracture point with a different local map, based on the relative surprisal of the different locations and the connectivity graph of maps in LTM (described below). Given sufficiently high surprisal at the current location, a fragmentation will occur. Upon fragmentation, the local map in STM is stored as an episode in LTM and then erased from STM. The STM begins a new local map. Stored local maps in LTM are recalled whenever the agent reaches a location where a fragmentation event had occurred. Section A and Algorithm 1 provide detailed procedures.

**Local Map** The STM has a local predictive spatial map, $\mathbf{M}^{\text{cur}}_t \in \mathbb{R}^{(C+1) \times H \times W}$ where the size $(H, W)$ grows as the agent extends its observations in the local region by adding newly discovered regions. The first $C$ channels of $\mathbf{M}^{\text{cur}}_t$ denote color and the last channel denotes the agent's confidence in each spatial cell. For simplicity, in this paper, we will focus only on the update of confidence channel ($C$-th channel). The local predictive map is simply a temporally decaying trace of recent sensory observations, shifted by the agent's actions (movement through space) (Klukas et al., 2021):

$$\mathbf{M}^{\text{cur}}_{t,C} = \gamma \cdot \mathbf{M}^{\text{cur}}_{t-1,C} + (1 - \gamma) \cdot o'_{t,C}, \tag{1}$$

where $\gamma$ is the decaying factor and $o_t \in \mathbb{R}^{(C+1) \times h \times w}$ is the egocentric view input observation in the environment at time $t$ sized as $(h, w)$. The last channel of the observation means visibility caused by occlusion or restricted FOV; visible (1) or invisible (0) on each cell. The red region is visible and others are invisible in Figure 2. $o'_t \in \mathbb{R}^{(C+1) \times H \times W}$ denotes a spatially transformed observation to $\mathbf{M}^{\text{cur}}_{t-1}$ to update the current observation to the local map in the correct position; rotation and zero-padding (Figure 8). Please refer to Section A for further details of how the local map is updated.

**Surprisal** The scalar surprisal signal $s_t = 1 - c_t$ is generated using the local map in STM and the current observation, where $c_t$ quantifies the average similarity of the visible part of observation to the local predictive map $\mathbf{M}_{t-1}^{\text{cur}}$ before update:

$$c_t = \frac{\mathbf{M}_{t-1,C}^{\text{cur}} \cdot o'_{t,C}}{||o'_{t,C}||_1}. \tag{2}$$

The agent is assumed to maintain a running estimate of the mean $\mu_t$ and standard deviation $\sigma_t$ of past surprisals, stored as part of the current map.

**Fragmentation** Fragmentation occurs if the $z$-scored current surprisal $((s_t - \mu_t)/\sigma_t)$ exceeds a threshold, $\rho$. Initially on each new map, the agent collects surprisal statistics and is not permitted to further fragment space until the number of samples is greater than 25 (for large sample condition). We also store the ratio $q_c$ of the number of frontier cells to the number of known cells and the distance between each fracture point in $\mathbf{M}_t^{\text{cur}}$. $q_c$ keeps updating every time step until it is stored in LTM. The ratio is used to guide agents on whether or not to move to other local maps. When $\mathbf{M}_t^{\text{cur}}$ is stored in LTM, it is connected with an adjacent map fragment that shares the same fracture point in the connectivity graph. In other words, the node of the graph is a model fragment and the connection denotes both fragments share a fracture point.

**Recall** Each local map records the fracture points. At these points, there are overlaps with other map fragments. When the agent moves to the point in the current local map, corresponding local map is recalled from LTM and the current one is stored in LTM.

**Subgoal** Subgoal is decided by using either the current local map in STM or the connectivity graph in LTM. The former enlarges the current local map while the latter helps find the next local map to explore. An agent explores the local region in the environment unless the current surprisal is too low (*e.g.*, $z$-score is smaller than $-1$) and there is less explored local map nearby.

Subgoals made with the current local map are based on frontier-based subgoals (Yamauchi, 1997) for exploring the local region. In the current local map, we first find all frontiers which are unknown cells adjacent to the known unoccupied cells. A group of consecutive frontiers are called 'frontier-edge' and Yamauchi (1997) uses the nearest centroid of the frontier-edge as a subgoal. Unlike standard SLAM methods that employ the entire map, our map in STM only covers a subregion of the environment. After fragmentation, the region, where the agent came from, has several frontiers (border of two local models) forming a frontier-edge. It leads the agent to go back to the previous area and recall the corresponding map fragment. This would lead to the agent moving between two map fragments for a long time. Hence, we prioritize the frontier-edge that is not located spatially behind the agent. The subgoal is sampled with the following weight $w_i$ for each frontier-edge $\mathcal{F}_i$:

$$w_i = \frac{|\mathcal{F}_i| \cdot \mathbb{1}(\mathcal{F}_i \text{ is not located spatially behind the agent})}{d_i}, \tag{3}$$

where $d_i$ means the distance between the current position and the centroid of $\mathcal{F}_i$ and $\mathbb{1}(\cdot)$ is the indicator function that is 1 if the condition is true otherwise 0.

Once the agent finishes exploring the local region, it should move to different subregions. However, subgoals from the current local map can misguide the already explored region since the agent does not have information beyond the map. Hence, we employ the connectivity graph of local maps stored in LTM. We leverage the discovery ratio (the ratio of the number of frontier cells to the number of known cells) $q$ mentioned above to find the most desirable subregions to explore. We also utilize the Manhattan distance between the current agent location and the fracture point between the current ($c$-th) local map and the connected $i$-th local map, $d_{i,c}$ where $d_{c,c} = 0$ and $d_{j,c} = \infty$ if $j$-th local map is disconnected to the current map. Then, the desirable local map is selected as

$$g = \arg\max_i \frac{q_i}{d_{i,c} + \epsilon}, \tag{4}$$

where $\epsilon$ denotes the preference of staying in the current local map; smaller value encourages staying in the current local map. If $g$ is not equal to $c$, the fracture point between the current local map and $g$-th local map is set to the subgoal. Once the agent arrives at the fracture point, the corresponding local map is recalled and the agent recursively checks Eq. 4 until $g$ is the arrived subregion. Note that the distances between a new location and other fracture points stored in the recalled local map are precomputed since they are fixed.

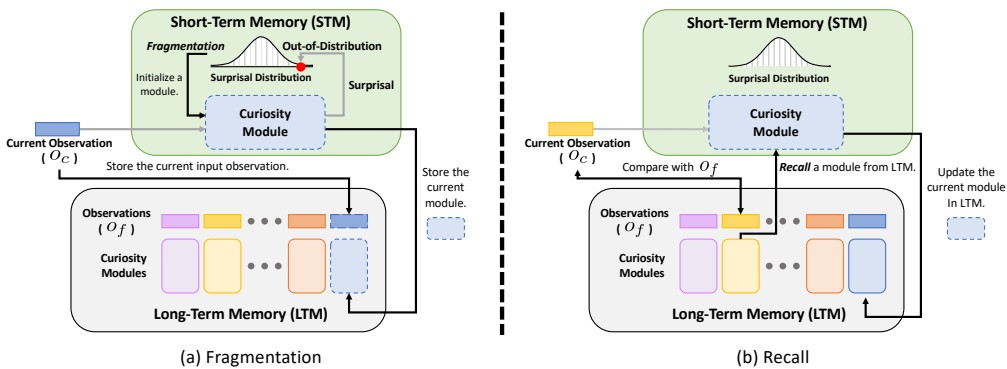

Figure 3: Illustration of the proposed FarCuriosity framework. (a) *Fragmentation*. If the current surprisal is out-of-distribution with high value (gray arrows), the current curiosity module and its input are stored in LTM and a new module is initialized. (b) *Recall*. If the current observation is similar to an observation in LTM, the corresponding curiosity module is recalled and we update the current module in LTM.

**Planner**   The planner takes a subgoal and the current spatial map in STM and finds the shortest-path within the map from the current agent location to the subgoal. We use Dijkstra's algorithm for planning a path to the next subgoal. However, the planner can be any path planning method such as A$^*$ algorithm (Hart et al., 1968) or RRT (LaValle, 1998).

## 3.2   FRAGMENTATION AND RECALL IN CURIOSITY-DRIVEN RL (FARCURIOSITY)

Like in spatial map building, exploration is a critical property of good reinforcement learning systems. Unlike map building, deep RL can be applied to diverse tasks including manipulation  (Lee et al., 2021; Tassa et al., 2018), game playing (Bellemare et al., 2013; Küttler et al., 2020) and also spatial foraging and navigation (Szot et al., 2021).

Intrinsic reward functions (in the form of curiosity modules) (Burda et al., 2019; Campero et al., 2021; Pathak et al., 2017; 2019; Raileanu & Rocktäschel, 2020) are used for exploring novel spaces of an environment by generating intrinsic rewards. The intrinsic reward can serve as the primary reward for training policies in realistic scenarios where external rewards are sparse or absent. Since a curiosity module is trained concurrently with the policy, it can suffer from catastrophic forgetting (French, 1999; Robins, 1995) over large and heterogeneous environments where the observation space is high variance such as having multiple rooms/stages with different backgrounds. Hence, demanding on larger model capacity to memorize what the model has seen for generating intrinsic rewards in these environments. The catastrophic forgetting of a curiosity module leads to high prediction errors (high intrinsic rewards) for already explored states, making agents less likely to explore unseen states.

We tackle this problem by applying our fragmentation-recall scheme to the problem of exploration in deep RL settings, implementing multiple local models in the form of multiple curiosity modules (*Far-Curiosity*). Similar to above, we now use surprisal to fragment the state space of an environment and agent. In *FarCuriosity*, surprisal is based on the predictions of a *curiosity module* in STM in place of the predictions using a local map in STM. While *FarMap* aims to efficiently build and use maps, the goal of *FarCuriosity* is to better explore states so that the agent can solve a given task.

**Fragmentation**   We use the prediction error generated by the curiosity module (Burda et al., 2019; Pathak et al., 2017) for fragmentation. When fragmentation occurs — a fracture point (high prediction error by the current curiosity module) — we store the current curiosity module and its input  (usually corresponding to the next observation) in LTM and initialize a new module.

**Recall**   At every point, the agent can search LTM to select a curiosity module. The curiosity module's input observation (or an encoded feature of the current observation) is the key used to select a module: Given the current input observation, $o_c$, and the observations from past fracture points, of which an element is $o_f$, and a fixed feature extractor $\phi(\cdot)$, we define a recall score $s_f^{\text{rec}}$ based on cosine similarity as follows:

$$s_f^{\text{rec}} = \frac{\phi(o_c) \cdot \phi(o_f)}{||\phi(o_c)||_2 ||\phi(o_f)||_2}.$$   (5)

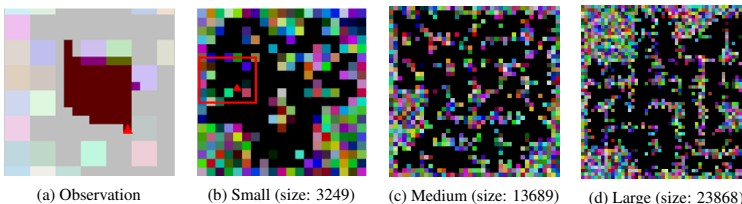

(a) Observation    (b) Small (size: 3249)    (c) Medium (size: 13689)    (d) Large (size: 23868)

Figure 4: Environments. Empty cells (that can be occupied by the agent) are black; walls are randomly colored. (a) Top-down visualization of the agent's local field of view (FOV) (agent: red triangle; shaded region: observation) within an environment (b). The agent has only a locally restricted egocentric view. The right side is occluded by a wall. (b) Top-down view of one environment. the red box marks the region shown in (a). (c), (d) Examples of medium and large environments.

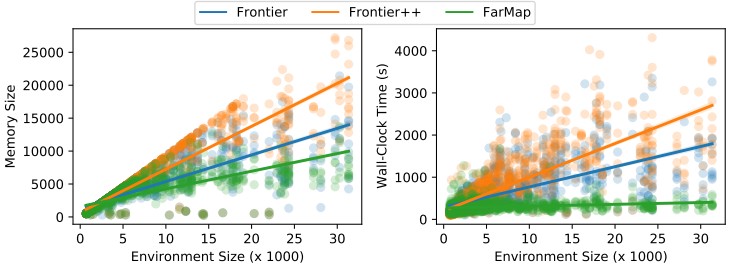

Figure 5: Relative memory and wall-clock time advantage of FarMap to Frontier-based baselines grows with environment size. Comparison of memory cost (left) and wall-clock time (right) as a function of environment size (circles: experimental results; line: linear regression fit). FarMap requires substantially less memory and is much faster than other methods.

If $s_f^{\text{rec}}$ is equal to or greater than a recall threshold $\psi$ for any element $o_f$, we store the current module in LTM and recall the module corresponding to $o_f$. We choose the highest scoring $f$ if multiple modules meet the condition. The module just before recall of the module for $o_f$ becomes a new element of the set of modules in LTM.

## 4 EXPERIMENTS

In this section, we conduct experiments for FarMap and FarCuriosity comparing their baselines on procedurally generated map environment and Atari games, respectively. Please refer to Section C for the experimental settings.

### 4.1 FARMAP EXPERIMENTS

We measure the map coverage and memory usage for each environment at each time step as our evaluation criteria and calculate the mean and standard deviation over all runs. The memory usage in each environment is calculated as a ratio of local map size (memory size, $H \times W$) to the environment size. Note that the local map size is the asymptotically dominant factor in the memory. We also use wall-clock time for comparison. We compare FarMap with standard frontier-based exploration (Frontier) (Yamauchi, 1997). We also consider an augmented baseline with heuristics used in FarMap: prioritizing the frontier-edges not behind the agent, stochastically selecting subgoals based on the number of frontiers and the distance to the centroid in each frontier-edge, mentioned in Section 3.1 (Frontier++). Our experiments are conducted in procedurally generated environments (See Section B). As shown in Figure 4, the walls in the environment are randomly colored and are composed of various narrow and wide pathways. For each trial, the agent is randomly placed before it begins to explore the environment. Figure 4a presents an example of the agent's view in the small environment shown in Figure 4b. The agent is presented as a red triangle and the observed cells are shaded. The agent has the restricted field of view with occlusion ($120°$).

Figure 5 and Table 1 analyze memory size and wall-clock-time changes depending on the environment size. In Table 1, we divide the environments into three groups; small, medium and large based on their sizes. FarMap requires significantly less memory than the baselines, with a small performance drop as the environment size is increased. Moreover, since our agent only refers to the subregion of the environment not using the entire map, it is much faster than other methods while planning. Especially,

Table 1: Comparison of average map coverage (%), memory use (%), and wall-clock time ($s$) for small, medium, and large environments. The memory usage advantage of FarMap relative to its counterparts grows with environment size. Frontier++ is Frontier (Yamauchi, 1997), augmented with head direction, and with a subgoal selection weights given by the size of the frontier in each subgoal. Although RND is much faster than others, it has much worse exploration performance. The numbers in parentheses are the standard deviation.

| Model | Small (size < 5,000) | | | Medium (5,000 ≤ size < 15,000) | | | Large (size ≥ 15,000) | | |
|---|---|---|---|---|---|---|---|---|---|
| | Coverage | Memory | Time | Coverage | Memory | Time | Coverage | Memory | Time |
| Frontier (Yamauchi, 1997) | 97.2 (9.2) | 80.4 (8.9) | 360.5 (168.8) | 76.3 (21.8) | 73.3 (19.7) | 871.9 (439.0) | 41.4 (20.1) | 44.4 (20.1) | 1261.0 (676.5) |
| Frontier ++ | 98.8 (6.7) | 81.6 (7.5) | 341.1 (191.1) | **91.4 (17.4)** | 85.9 (15.2) | 1099.5 (511.3) | **60.8 (21.4)** | 68.1 (20.9) | 1872.4 (782.7) |
| FarMap | **99.0 (7.2)** | **79.1 (8.7)** | **278.2 (118.6)** | 86.4 (19.8) | **62.9 (19.3)** | **321.4 (119.0)** | 56.6 (20.8) | **31.4 (11.8)** | **352.5 (110.7)** |
| RND (Burda et al., 2019) | 73.7 (23.3) | - | 50.6 (22.4) | 32.2 (16.2) | - | 48.8 (16.2) | 13.3 (7.6) | - | 52.7 (43.6) |

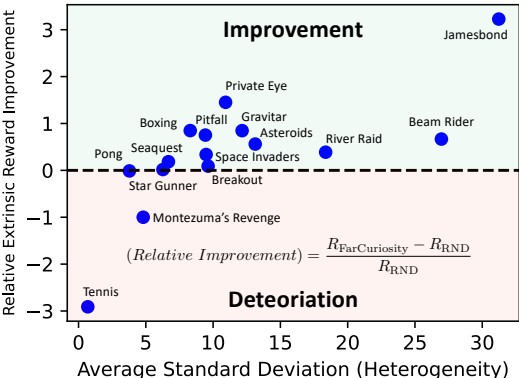

Figure 6: The relative performance improvement over RND performance ($R_{\text{RND}}$) in various Atari games. The average standard deviation is measured from each pixel in a trajectory generated by trained RND. FarCuriosity is effective in visually high-variance (heterogeneous) games.

in large environments, it is approximately four times faster than the baseline with heuristics. We also compare with Random Network Distillation (RND) (Burda et al., 2019) trained and evaluated on each map separately without extrinsic reward. Although RND requires constant time for any size of environment, its exploration performance is worse than other methods. We additionally show how map coverage and memory usage change in exploration in Section D, conduct sensitivity analysis of hyperparameters in Section E, and confidence interval of Table 1 in Section F.

## 4.2 FARCURIOSITY EXPERIMENTS

We present the empirical results that verify the problem of catastrophic forgetting (Section 3.2) of intrinsic rewards in curiosity-driven deep reinforcement learning in this section. We implement our FarCuriosity method based on RND (Burda et al., 2019) and compare against to the vanilla RND. Both FarCuriosity and the vanilla RND train a policy using Proximal Policy Optimization (PPO) (Schulman et al., 2017), as Burda et al. (2019) suggest. We follow the original RND paper and select Atari (Bellemare et al., 2013) as the benchmark. We perform experiments over 15 Atari games: Asteroids, Beam Rider, Boxing, Breakout, Gravitar, Jamesbond, Montezuma's Revenge, River Raid, Pitfall, Pong, Private Eye, Seaquest, Space Invaders, Star Gunner, and Tennis.

We measure the average across all pixels of the standard deviation of each pixel from trajectories generated by trained RND (Burda et al., 2019) as the *heterogeneity* of each game. If the agent only explores a small portion of the environment, the fragmentation is not needed although the environment has hundreds of distinctive rooms. Therefore, we measure heterogeneity using our curiosity module, RND. Figure 6 shows the relative performance improvement on multiple Atari games which is defined as below:

$$(Relative\ Improvement) = \frac{R_{\text{FarCuriosity}} - R_{\text{RND}}}{R_{\text{RND}}}, \qquad (6)$$

where $R_{\text{FarCuriosity}}$ and $R_{\text{RND}}$ denote the average extrinsic reward of FarCuriosity and RND, respectively. Since both $R_{\text{FarCuriosity}}$ and $R_{\text{RND}}$ are negative in Tennis and Pitfall, we multiply the improvement by -1 here. The negative improvement denotes the performance is worsened by FarCuriosity. FarCuriosity generally improves performance in heterogeneous environments while achieving similar or even worse performance in homogeneous games. FarCuriosity achieves worse performance

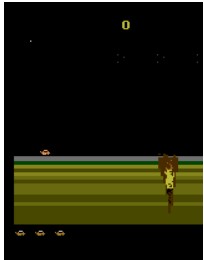

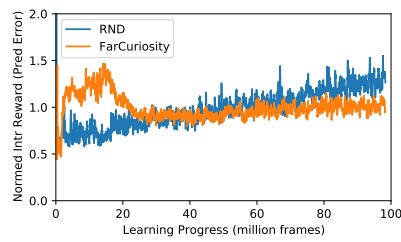

(a) Starting observation                    (b) Intrinsic reward of starting observation

Figure 7: Intrinsic reward of the same observation (first observation in every episode) in Jamesbond during training. Each reward is normalized by the average intrinsic reward across training separately, in order to directly compare the magnitude of the change in intrinsic rewards during training. The reward from RND is increasing while one from FarCuriosity is decreasing after 15 millions steps.

in Montezuma's Revenge which might be due to its low heterogeneity since RND could not explore many rooms. FarCuriosity has the worst deterioration in the least heterogeneous environment, Tennis. We analyze the relationship between the number of fragments and the performance in Section G and the experiments in each environment in Section H.

FarCuriosity does significantly better than RND in Jamesbond; to explain this, we measure the intrinsic reward (prediction error) when the agent returns to the same observation (first frame of the episode) during training as shown in Figure 7. The error of the FarCuriosity agent at the observation remains the same after a short initial increase, regardless of how much subsequent learning the agent performs elsewhere in the environment. On the other hand, RND generates a higher intrinsic reward (higher prediction error) as training progresses, implying that this method suffers from catastrophic forgetting. The prediction-based curiosity module assumes that infrequently visited states have high prediction error and vice versa. However, due to catastrophic forgetting, this assumption is invalid here. In summary, our *fragmentation-and-recall* concept is helpful for heterogeneous environments.

## 5 DISCUSSION

We proposed a new framework for exploration based on local models and fragmentation, inspired by how natural agents explore space. Our framework *fragments* the exploration space based on the current surprisal in real time and stores the current model fragment in long-term memory (LTM). Stored fragments are *recalled* when the agent returns to the state where the fragmentation happened so that the agent can reuse the local information. Accordingly, the agent can refer to *longer-term* local information. We believe that the framework can be applicable to any tasks that use streaming observations or data which are reused or recurring. We applied this framework to the settings of spatial exploration (*FarMap*) and general reinforcement learning exploration (*FarCuriosity*). The surprisal is generated by short-term memory (STM) using a local map in FarMap and a curiosity module in FarCuriosity. Consequently, FarMap requires less memory and wall-clock time than the baseline method (Yamauchi, 1997) without sacrificing exploration performance. On the other hand, FarCuriosity learns better than its baseline (Burda et al., 2019) in standard deep reinforcement learning benchmarks by appropriately recalling model fragments once fragmentation happens in the environment. In principle, as in *FarMap*, LTM could include a connectivity graph that describes the transition structure between recalled modules, which we believe will be useful for goal-directed reinforcement learning. However, in what follows, we use only individual stored modules in LTM for FarCuriosity and not the connectivity graph, which we leave for future work.

Our paper aims to be a proof-of-concept for fragmentation and recall; we have applied it to only two domains with a relatively small number of environments, but the strong performance in these cases suggests that the concept should provide improvements in many heterogeneous-environment learning problems. On the contrary, FarCuriosity is *not* effective in homogenous environments or frequently reset environments since the baseline agent does not suffer from catastrophic forgetting in those environments. With this effort, we intend to bring to the attention of the machine learning community the *fragmentation-and-recall* concept inspired by natural agents for heterogeneous environments. This concept can make large-scale exploration, which typically requires a huge memory size and long-ranged memory span, significantly more efficient.

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

APPENDIX

---

**Algorithm 1** FarMap Procedure at time $t$. FarMap algorithm is colored in red and heuristics are colored in blue on top of Frontier algorithm.

---

**Require:** a spatial map $\mathbf{M}_{t-1}^{\text{curr}}$, previous action $a_{t-1}$, current observation $o_t$, short-term memory STM, long-term memory LTM
**Ensure:** Updated map, $\mathbf{M}_t^{\text{curr}}$ and a sequence of actions $\{a\}$

1: **procedure** STEP
2:      $\mathbf{M}_t^{\text{curr}} = \gamma \cdot \mathbf{M}_{t-1}^{\text{curr}} + (1 - \gamma) \cdot o_t'$          ▷ Update the current local map
3:      Calculate $s_t = 1 - c_t$ following Eq. 2.
4:      **if** current position in $\mathbf{M}_t^{\text{curr}}$ is fracture point **then**          ▷ Recall
5:          $q_c = N_{\text{frontier}}$ / $N_{\text{known}}$
6:          Store $\mathbf{M}_t^{\text{curr}}$ in LTM.
7:          Recall a corresponding map fragment in LTM to STM (*i.e.*, change $\mathbf{M}_t^{\text{curr}}$)
8:      **else if** $z_t > \rho$ **then**          ▷ Fragmentation
9:          $q_c = N_{\text{frontier}}$ / $N_{\text{known}}$
10:         Store $\mathbf{M}_t^{\text{curr}}$ in LTM.
11:         Initialize a new map $\mathbf{M}_t^{\text{curr}}$ in STM.
12:      **end if**
13:      Update $\mu_t$ and $\sigma_t$.
14:      $g = \arg\max_i \frac{q_i}{d_{i,c} + \epsilon}$
15:      **if** $g \neq c$ **then**          ▷ Subgoal based on connectivity between fragments.
16:         *subgoal* ← the fragmentation location between current fragment $c$ and a fragment $g$
17:      **else**
18:         Find frontier-edges $\{\mathcal{F}_i\}$ and their centroids.
19:         Calculate distance between the current position and each centroid $\{d_i\}$.
20:         $w_i = 1/d_i \cdot |\mathcal{F}_i| \cdot \mathbb{1}(\mathcal{F}_i \text{ is not located spatially behind the agent})$    ▷ $\mathbb{1}(\cdot)$ is 1 if the condition is true else 0.
21:         Select frontier-edge $\mathcal{F}_g$ based on the weighted sampling with $\{w_i\}$.
22:         *subgoal* ← the nearest frontier $\in \mathcal{F}_g$ from its centroid.
23:      **end if**
24:      A sequence of actions, $\{a\}$ ← Planner(*subgoal*; $\mathbf{M}_t^{\text{curr}}$)          ▷ Dijkstra's algorithm.
25: **end procedure**

---

## A    OVERALL PROCEDURE OF SPATIAL NAVIGATION

Algorithm 1 presents the overall procedure of FarMap at time $t$. On top of Frontier algorithm (Yamauchi, 1997), we colored heuristics that is used for Frontier++ and FarMap as blue (Line 20) and FarMap algorithm as red. Given the previous action $a_{t-1}$, current observation $o_t$, a local predictive map $\mathbf{M}_{t-1}^{\text{curr}}$, we first update the map following Eq. 1 and calculate the surprisal $s_t$ following Eq. 2. Figure 8 shows the toy illustration of how to transform the current observation to update the local map and how the map size grows. We first rotate the observation following the head direction of the agent in the map and then zero-pad it so that it has the same size as the local map considering the agent's current location in the map. If the observation does not fit in the same size of the map due to the agent's location, we add zero-padding (gray in the figure) to both the transformed observation and the local map. Then, we update the local map by adding the transformed observation.

If the current position in $\mathbf{M}_t^{\text{curr}}$ is previously fragmented location with another fragment, we calculate $q_c$ as the number of frontiers ($N_{\text{frontier}}$) over the number of known cells ($N_{\text{known}}$) in the map $\mathbf{M}_t^{\text{curr}}$ (Line 5). Then, we store the current map in LTM and recall the corresponding map fragment in LTM to STM. If recall is not happened and $z_t$ is greater than a threshold $\rho$, we store the current map in LTM and initialize a new map in STM.

After checking recall and fragmentation, we find the desirable local map fragment that is less explored compared to other fragments mentioned in Section 3.1. If the current map is not the desirable map, we set subgoal as the fracture point between the current map and the desirable map. Otherwise, we

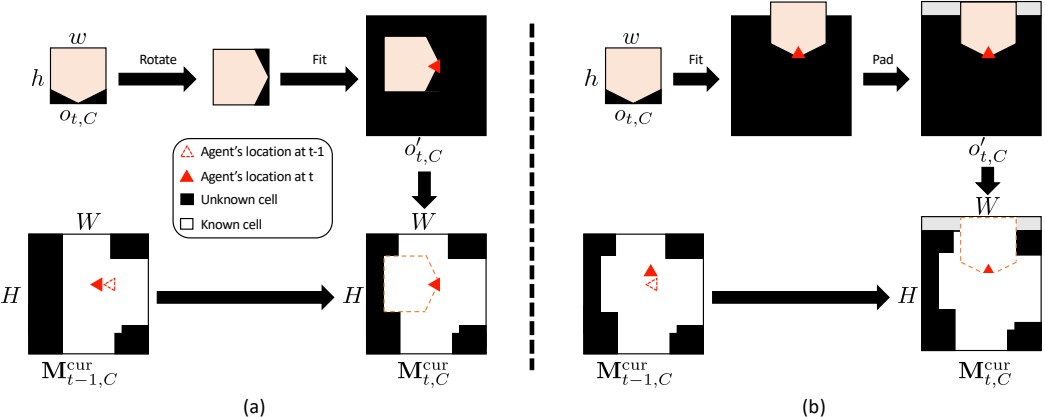

Figure 8: Toy example illustrations of how the local map is updated. In this figure, we only consider the visibility of each cell without considering occupancy and the color for simplification. (a) We first rotate the current observation $o_{t,C}$ based on the head direction of the agent in the local map. Then, the observation is zero-padded to have the same size as the local map. Finally, the local map is updated by adding transformed observation $o'_{t,C}$. (b) If the current observation does not fit in the local map due to the agent's location, we add zero-padding (gray) to both observation and the local map. Hence, the size of the local map has increased ($H$ has changed).

first find frontier-edges and calculate a weight of each frontier-edge $\mathcal{F}_i$ for weighted sampling with weight $w_i$ following Eq. 3 ($w_i$ is $1/d_i$ in Frontier model).

The subgoal is defined as the nearest frontier from the centroid of sampled frontier-edge. Finally, a planner generates a sequence of actions to go to the subgoal. Note that while the agent moves based on the sequence, it keeps update the map and checks fragmentation and recall.

## B   PROCEDURALLY-GENERATED ENVIRONMENTS

We use the procedurally-generated environment for the map building experiments in Section 4 in the main paper. Figure 9 and Algorithm 2 show the procedure of map generation. We first generate grid patterned square rooms and randomly connect and merge them. Then, we flip boundary cells of empty or occupied multiple times for diversity. Formally, given the length of square $S$, the interval between square rooms, $L$, and the size of the grid, $(N, M)$, we first generate the binary square grid map $\mathcal{M} \in \{0 \text{ (empty)}, 1 \text{ (occupied)}\}^{(N \cdot S + (N+1) \cdot L) \times (M \cdot S + (M+1) \cdot L)}$ (Line 2 in Algorithm 2). Let $s_i$ be the $i$-th square as a row-major order in $\mathcal{M}$. For each of adjacent square pairs, we connect two squares with probability $p_{\text{connect}}$ as a width $w \sim \text{unif}\{1, 2, \ldots, S-1\}$ (Line 7) or merge them with probability $p_{\text{merge}}$ (Line 11). Then, we flip all boundaries between occupied and empty cells $K$ times with probability $p_{\text{flip}}$ (Line 17). After flipping the boundaries, there are several isolated (i.e. not connected to other submaps) submaps in $\mathcal{M}$. We only use the submaps where the sizes are greater than a threshold ($3S^2$ in our implementation) (Line 20). After creating maps, we randomly colorize each occupied cell and scale up by factor of 3. Note that the proposed environment has very complex maps compared to existing environments (Chevalier-Boisvert et al., 2018; Küttler et al., 2020)

## C   EXPERIMENTAL DETAILS

Our models are implemented on PyTorch and the experiments are conducted on Intel(R) Xeon(R) CPU E5-2650 v4 @ 2.20GHz for spatial exploration experiments and a single NVIDIA Tesla V100 GPU for reinforcement learning experiments. We will release the entire codebase once our paper gets accepted. However, we attached the code for environment generation for FarMap experiments.

---

**Algorithm 2** Spatial Exploration Environment Generation

**Require:** $N, M, L, S, K, p_{\text{connect}}, p_{\text{merge}}, p_{\text{flip}}$
**Ensure:** A set of maps, $\{\mathcal{M}\}$.
 1: **procedure** MAPGENERATION
 2:     Initialize $\mathcal{M} \in \{0,1\}^{(N \cdot S + (N+1) \cdot L) \times (M \cdot S + (M+1) \cdot L)}$, $(N, M)$ grid with interval $L$ and each square sized $(S, S)$.         ▷ Figure 9 (1).
 3:     **for** $(s_i, s_j) \in \{(s_i, s_j)|s_i \text{ and } s_j \text{ are adjacent}, i \leq j\}$ **do**    ▷ Get adjacent grid square pairs.
 4:         $x \sim \mathcal{B}(1, p_{\text{connect}})$         ▷ Connect adjacent squares with probability $p_{\text{connect}}$.
 5:         **if** $x = 1$ **then**
 6:             $w \sim \text{unif}\{1, \ldots, S - 1\}$
 7:             Connect $s_i$ and $s_j$ with width $w$.         ▷ Figure 9 (2).
 8:         **end if**
 9:         $x \sim \mathcal{B}(1, p_{\text{merge}})$         ▷ Merge adjacent squares with probability $p_{\text{merge}}$.
10:         **if** $x = 1$ **then**
11:             Merge $s_i$ and $s_j$ by removing the interval.         ▷ Figure 9 (3).
12:         **end if**
13:     **end for**
14:     **for** $k \leftarrow 1 \text{ to } K$ **do**
15:         **for** $c \in \{c | c \in \mathcal{M}, \exists_{c'} c \text{ xor } c' = 1, c' \in \text{Adj}(c)\}$ **do**    ▷ Get boundary cells in the map.
16:             $x \sim \mathcal{B}(1, p_{\text{flip}})$         ▷ Flip the cell with probability $p_{\text{merge}}$.
17:             $c = c \text{ xor } x$         ▷ Figure 9 (4)-(6).
18:         **end for**
19:     **end for**
20:     Divide $\mathcal{M}$ into a set of isolated maps $\{\mathbf{m}_i\}$         ▷ Figure 9 (7).
21:     Filter out a map in $\{\mathbf{m}_i\}$, where the size is smaller than $3S^2$.
22:     Randomly colorize the occupied cell in each map.         ▷ Figure 9 (8).
23:     Scale up each map in $\{\mathbf{m}_i\}$ by factor of X.
24: **end procedure**

---

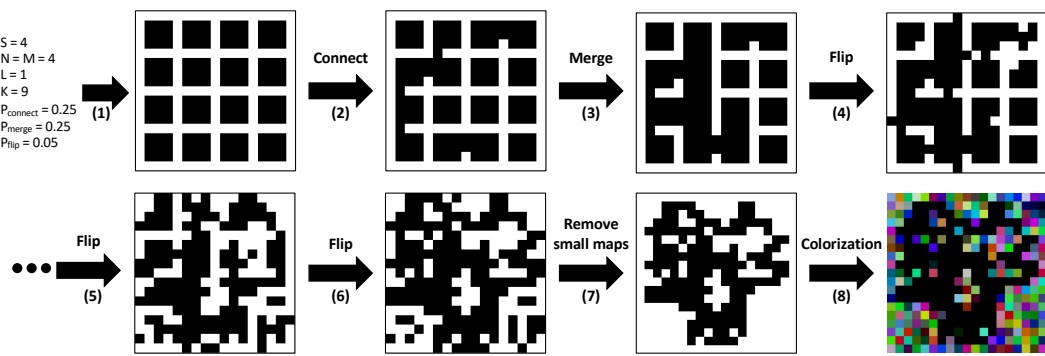

Figure 9: Procedure of map generation. (1) We first set square grid where black and white denote empty and occupied, respectively. (2) We randomly connect and (3) merge adjacent grid. (4)-(6) We also randomly flip the boundaries of empty and occupied cell recursively. (7) Then, we remove small isolated subregions and (8) randomly colorize occupied cells. Finally, we increase the size of the map.

## C.1 FARMAP ENVIRONMENT GENERATION

To generate the environment, we run map generation (Algorithm 2) 200 times and then use the 300 largest sized maps. All maps are scaled up by a factor of 3 after colorization for the task. On every trial, we sample $S$ and $N$ from $\{3, 4, 5, 6, 7\}$ and set $M = N$. $K, L \in \mathbb{N}$ are sampled from $[0, 10]$ and $[1, 3]$, respectively. We set $p_{\text{connect}}$, $p_{\text{merge}}$ and $p_{\text{flip}}$ to $0.25, 0.25, 0.05$, respectively.

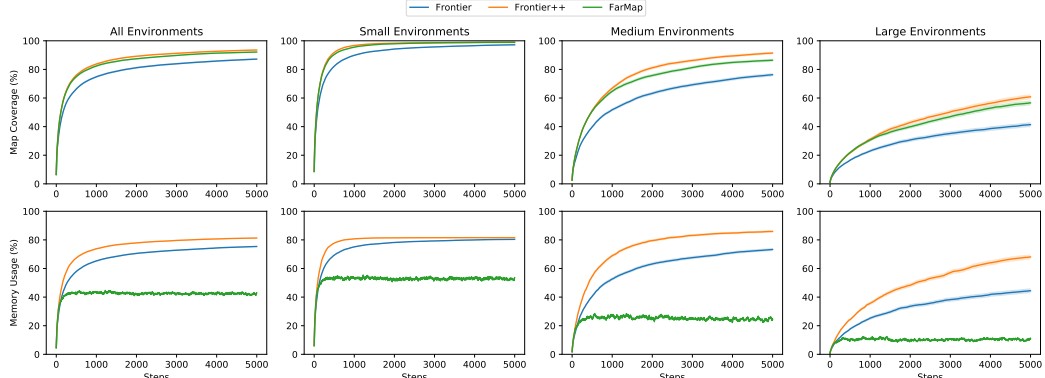

Figure 10: Growth in agent-explored map region as a function of number of steps from the first step in the environment matches the performance of an augmented Frontier-based baseline with less memory use. Mean spatial map coverage performance (up) and mean memory usage (down) as a function of number of steps taken in various sizes of environment sets. FarMap achieves better or comparable exploration as a Frontier-based exploration baseline (Frontier) (Yamauchi, 1997) and an improved version of it (Frontier++) while using only about half the memory on average. The memory benefit is increased as in larger environment.

## C.2 FARMAP

We run the agent on 1500 different environments: 300 different maps with five random seeds and the starting position and the color of map is changed on each random seed. We set $\gamma$, $\rho$ and $\epsilon$ to 0.9, 2 and 5, respectively. The observation size $(h, w)$ is (15,15). If the frontier-based exploring agent is surrounded by a large frontier-edge in an open space, the centroid of the frontier can fall in the interior of the explored space, leading to no new discovery. This causes the agent to become stuck. We improve the agent by instead selecting the nearest unoccupied cell from the nearest frontier state from the centroid. We train RND for 1 million steps without extrinsic reward for each environment with five different random seeds, and other parameters are the same in Section C.3.

## C.3 FARCURIOSITY

We run Atari experiments for 100 million frames, and we optimize in batches of 128 frames per environment across 32 parallel environments with minibatch size, 4. The learning rate is 0.0001, the reward discount factor $\gamma$ is 0.99 and the number of epochs is 4. For other parameters, we use the same values mentioned in PPO and RND: we set GAE parameter $\lambda$ as 0.95, value loss coefficient as 1.0, entropy loss coefficient as 0.001, and clip ratio ($\epsilon$ in Eq. (7) in Schulman et al. (2017)) as 0.1. We set the fragmentation parameter $\rho$ to 10, and the recall parameter $\psi$ is to 0.99. The maximum size of the LTM is 800, and the least recently used fragment is substituted by a new fragment if LTM is fully used. For fragmentation, we also add a cosine similarity constraint: if the cosine similarity $s_f$ with any observation in the fragmentation moment is higher than 0.75, we skip the fragmentation to avoid excessive fragmentation. We use a fixed random network in RND as a feature extractor $\phi(\cdot)$ for Eq. 5 and all curiosity modules share the same fixed network.

## D COMPARISON ON EACH STEP IN FARMAP EXPERIMENTS

Figures 10 summarizes the experimental results on 1500 environments on each step. The lines in the plots are the average of all or a group of experiments and the shaded areas are standard errors of the mean which are not visible due to large number of trials. In general, FarMap uses stable amount of memory on average (40 %) over all exploration while other methods use much more memory as map coverage increases. The average memory usage of FarMap is almost consistent in any group of environments as the agent explores environments while the usages of Frontier and Frontier++ keep increasing. Moreover, as we already mentioned in Section 4.1, the memory usage gap between FarMap and Frontier++ is dramatically increased while the map coverage gap is small.

Table 2: Sensitivity analysis about fragmentation threshold, $\rho$ in FarMap.The numbers in parentheses are the standard deviation.

| $\rho$ | Small (size < 5,000) | | | Medium (5,000 ≤ size < 15,000) | | | Large (size ≥ 15,000) | | |
|---|---|---|---|---|---|---|---|---|---|
| | Coverage | Memory | Time | Coverage | Memory | Time | Coverage | Memory | Time |
| 1.0 | **99.1 (7.0)** | **71.5 (11.3)** | **117.9 (34.1)** | 87.1 (19.4) | **39.6 (13.7)** | **146.2 (61.5)** | **60.9 (19.8)** | 17.9 (6.6) | **148.4 (36.1)** |
| 1.5 | 99.1 (7.1) | 75.7 (9.9) | 158.0 (653.5) | 87.6 (19.0) | 50.2 (17.2) | 180.1 (60.3) | 59.7 (20.1) | 23.3 (9.3) | 188.9 (51.3) |
| 2.0 (ours) | 99.0 (7.2) | 79.1 (8.7) | 278.2 (118.6) | 86.4 (19.8) | 62.9 (19.3) | 321.4 (119.0) | 56.6 (20.8) | 31.4 (11.8) | 352.5 (110.7) |
| 2.5 | 98.8 (6.9) | 80.7 (8.1) | 207.1 (503.4) | 89.0 (18.2) | 79.7 (18.0) | 557.3 (336.8) | 58.4 (20.3) | 56.9 (19.0) | 770.5 (393.0) |
| 3.0 | 98.8 (6.7) | 81.5 (7.6) | 296.1 (154.5) | **91.0 (17.5)** | 85.0 (15.6) | 698.2 (308.8) | 60.9 (21.5) | 67.9 (21.0) | 1068.0 (413.1) |

Table 3: Sensitivity analysis about decaying factor, $\gamma$ in Eq. 1 in FarMap. The numbers in parentheses are the standard deviation.

| $\gamma$ | Small (size < 5,000) | | | Medium (5,000 ≤ size < 15,000) | | | Large (size ≥ 15,000) | | |
|---|---|---|---|---|---|---|---|---|---|
| | Coverage | Memory | Time | Coverage | Memory | Time | Coverage | Memory | Time |
| 0.8 | 98.8 (7.2) | 79.4 (8.5) | 210.5 (104.2) | 85.3 (20.6) | 64.5 (20.7) | **304.0 (228.4)** | 55.6 (19.7) | 32.8 (11.3) | 304.8 (112.1) |
| 0.9 (ours) | 99.0 (7.2) | 79.1 (8.7) | 278.2 (118.6) | 86.4 (19.8) | 62.9 (19.3) | 321.4 (119.0) | 56.6 (20.8) | **31.4 (11.8)** | 352.5 (110.7) |
| 0.95 | **99.1 (6.8)** | **79.0 (8.4)** | **178.3 (74.3)** | 87.3 (19.1) | **61.3 (19.1)** | 507.5 (4564.6) | 59.2 (20.2) | 31.9 (12.5) | **284.7 (87.7)** |
| 0.99 | 99.1 (6.8) | 80.8 (7.9) | 262.3 (232.4) | **89.3 (18.3)** | 76.5 (18.3) | 453.8 (210.3) | **60.4 (20.2)** | 46.7 (18.4) | 541.5 (231.6) |

# E  SENSITIVITY ANALYSIS FOR HYPERPARAMETERS IN FARMAP

We test FarMap with various hyperparameters; fragmentation threshold ($\rho$), decaying factor ($\gamma$), and $\epsilon$. All experiments are conducted in the same environments. While comparing one hyperparamters, we fix the remaining parameters as $\rho = 2.0, \gamma = 0.9, \epsilon = 5$. Tables 2 presents the performance of FarMap with different fragmentation threshold, $\rho$. The smaller value makes it more prone to fragment the space, which means it can use less memory but it overly fragments the space. On the other hand, bigger threshold makes use more memory without fragmentation. Hence, we choose 2 as the threshold value (95% confidence interval if the distribution follows gaussian). On the other hand, our FarMap is robust to the decaying factor and $\epsilon$ as shown in Table 3 and Table 4, respectively.

# F  STATISTICAL ANALYSIS OF FARMAP EXPEIRMENTS

Table 5 shows a 95 % confidence interval by using bootstrapping with one million samples on FarMap experiments. The confidence interval is very wide since our metrics map coverage, memory usage, and wall-clock time depend on the size and the complexity of the environment and each method is evaluated on many varied environments. To reduce the effect of variance in environments, we present the relative results comparing to Frontier (dividing the results of Frontier++ and FarMap by the result of Frontier) in Table 6 and Figure 11. For Figure 11, we first sort the environments based on their sizes, and then we partition the environments into groups of size 50 and calculate the average and run bootstrapping to get a 95 % confidence interval for each group. The confidence intervals of map coverage for both Frontier++ and FarMap are similar. On the other hand, memory usage and wall-clock time are significantly different although there are some overlaps.

# G  EFFECT OF THE NUMBER OF FRAGMENTATION ON FARCURIOSITY PERFORMANCE

Figure 12 shows the relation between the relative performance improvement and the average number of fragments on multiple Atari games. In most games, the number of fragments is up to 50, but it reaches the maximum capacity of LTM in Beam Rider. There is no clear global tendency between performance improvement and the number of fragments.

Table 4: Sensitivity analysis about $\epsilon$ in Eq. 4 in FarMap. The numbers in parentheses are the standard deviation.

| $\epsilon$ | Small (size < 5,000) | | | Medium (5,000 ≤ size < 15,000) | | | Large (size ≥ 15,000) | | |
|---|---|---|---|---|---|---|---|---|---|
| | Coverage | Memory | Time | Coverage | Memory | Time | Coverage | Memory | Time |
| 1 | 99.0 (7.1) | 79.1 (8.7) | 198.0 (86.4) | **86.8 (19.4)** | 63.0 (19.2) | 275.3 (93.7) | 56.6 (20.7) | 31.5 (11.9) | 294.8 (81.4) |
| 3 | **99.0 (7.2)** | **79.1 (8.7)** | 198.1 (88.3) | 86.7 (19.5) | 63.0 (19.1) | **271.3 (89.1)** | 56.5 (20.7) | 31.5 (11.9) | **294.5 (81.5)** |
| 5 (ours) | 99.0 (7.2) | 79.1 (8.7) | 278.2 (118.6) | 86.4 (19.8) | **62.9 (19.3)** | 321.4 (119.0) | **56.6 (20.8)** | 31.4 (11.8) | 352.5 (110.7) |
| 10 | 99.0 (7.2) | 79.1 (8.7) | **197.1 (91.9)** | 86.6 (19.5) | 62.9 (19.1) | 272.5 (88.2) | 56.3 (20.9) | 31.4 (11.8) | 294.9 (82.1) |
| 15 | 99.0 (7.3) | 79.1 (8.7) | 198.5 (94.4) | 86.6 (19.5) | 63.0 (19.1) | 288.7 (247.9) | 55.9 (20.7) | **31.1 (11.6)** | 295.7 (84.4) |

Table 5: Comparison of average map coverage (%), memory use (%), and wall-clock time ($s$) for small, medium, and large environments. The memory usage advantage of FarMap relative to its counterparts grows with environment size. The numbers in parentheses are 95 % confidence interval generated by bootstrap with one million samples.

| Model | Small (size < 5,000) | | | Medium (5,000 ≤ size < 15,000) | | | Large (size ≥ 15,000) | | |
|---|---|---|---|---|---|---|---|---|---|
| | Coverage | Memory | Time | Coverage | Memory | Time | Coverage | Memory | Time |
| Frontier | 97.2 (76.0, 100.0) | 80.4 (61.8, 88.7) | 360.5 (154, 773) | 76.3 (15.6, 99.8) | 73.3 (13.0, 92.3) | 871.9 (290, 2020) | 41.4 (6.1, 84.3) | 44.4 (3.8, 84.3) | 1261.0 (217, 3189) |
| Frontier ++ | 98.8 (97.5, 100.0) | 81.6 (71.1, 88.7) | 341.1 (118, 852) | **91.4 (15.2, 99.8)** | 85.9 (12.5, 93.3) | 1099.5 (365, 2339) | **60.8 (6.1, 98.7)** | 68.1 (3.8, 94.2) | 1872.4 (421, 3785) |
| FarMap | **99.0 (96.3, 100.0)** | 79.1 (61.4, 88.0) | 278.2 (139, 538) | 86.4 (15.6, 100.0) | **62.9 (12.5, 90.2)** | **321.4 (191, 528)** | 56.6 (6.1, 97.7) | **31.4 (3.8, 54.3)** | **352.5 (202, 633)** |

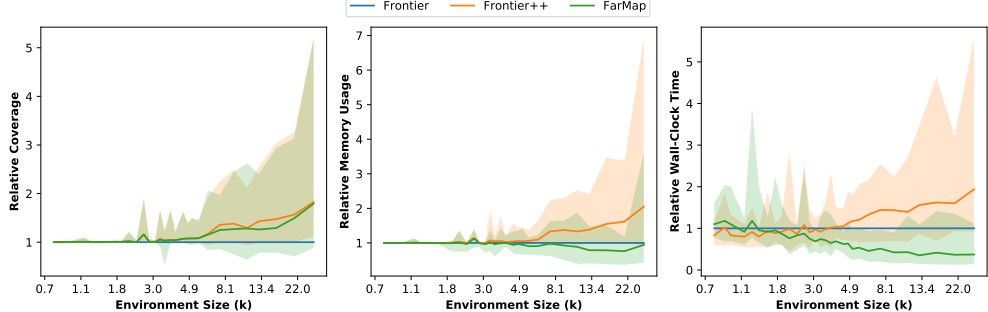

Figure 11: Relative map coverage, memory usage, and wall-clock time normalized by the results from Frontier on each environment. The mean (line) and 95% confidence interval (shade) are calculated by bootstrap with one million samples each from groups of 50 environments ordered by size.

## H  INDIVIDUAL PERFORMANCE IN ATARI GAMES

Figure 13 presents the mean cumulative extrinsic rewards (also known as extrinsic return) and its standard errors at varying number of training data (frames) over three different random trials in fifteen games including four hard Atari games: Gravitar, Montezuma's Revenge, Pitfall and Private Eye. FarCuriosity outperforms RND in ten environments and has worse performance in two environments. In Breakout and Space Invaders, FarCuriosity outperforms RND by a smaller margin. This is because the observations in those environments are more similar, and once the agent kills all enemies or hits all of the bricks, the same enemies and bricks are revived. Hence, RND does not suffer from catastrophic forgetting and achieves similar results with FarCuriosity. FarCuriosity shows similar performance in Pong, Star Gunner, and Pitfall. We believe this is because the observation space is less visually diverse in those games. In Pong for example, the only visually dynamic features are the agent, the opponent, and the ball, all of which move over a fixed monotone background. Moreover, no fragmentations occur in Star Gunner. Consequently, catastrophic forgetting does not happen and FarCuriosity does not demonstrate a significant advantage in this environment. On the other hand, FarCuriosity shows worse performance compared to RND in Tennis and Montezuma's Revenge. Tennis is the most homogeneous game in Atari as shown in Figure 6 and Montezuma's Revenge is also one of the most homogeneous games according to our measurement since RND agent is difficult to explore many rooms. We hypothesize that this homogeneity can lead to wrong fragmentation, which degrades the performance.

Table 6: Average and 95% confidence interval of relative map coverage, memory usage and wall-clock time of Frontier++ and FarMap over Frontier. Confidence interval is calculated by bootstrap with one million samples.

| Model | Small (size < 5,000) | | | Medium (5,000 ≤ size < 15,000) | | | Large (size ≥ 15,000) | | |
|---|---|---|---|---|---|---|---|---|---|
| | Coverage | Memory | Time | Coverage | Memory | Time | Coverage | Memory | Time |
| Frontier++ | 1.02 (0.99, 1.20) | 1.03 (1.10, 1.20) | 0.95 (0.56, 1.60) | **1.26 (0.99, 2.30)** | 1.24 (0.97, 2.20) | 1.36 (0.68, 2.70) | **1.65 (1.00, 3.30)** | 1.78 (1.00, 3.80) | 1.75 (0.80, 4.80) |
| FarMap | **1.03 (0.99, 1.20)** | 1.00 (0.72, 1.20) | **0.85 (0.41, 1.70)** | 1.20 (0.79, 2.20) | **0.90 (0.42, 1.60)** | **0.46 (0.16, 1.30)** | 1.56 (0.79, 3.20) | **0.84 (0.37, 1.70)** | **0.37 (0.13, 1.30)** |

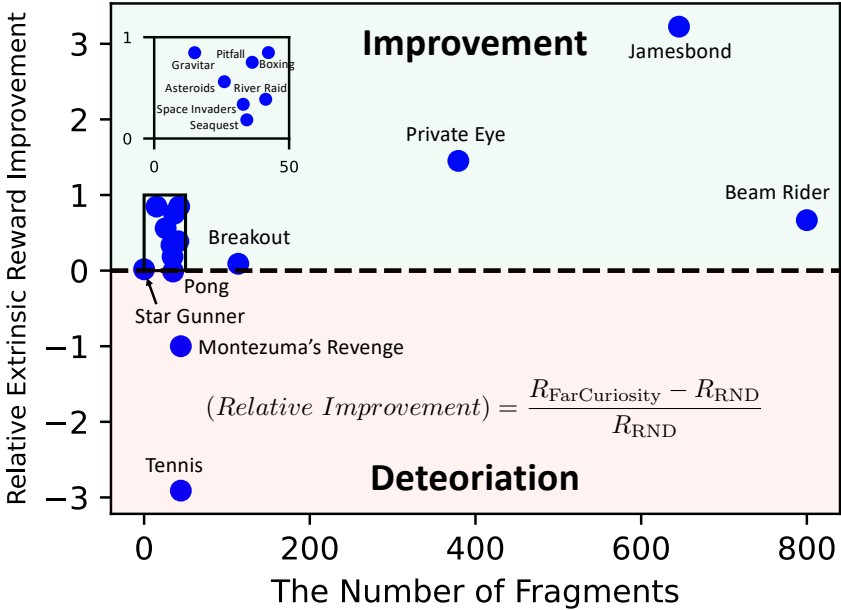

Figure 12: The relative performance improvement over RND performance ($R_{\mathrm{RND}}$) in various Atari games. The average number of fragments is at most 50 in most games.

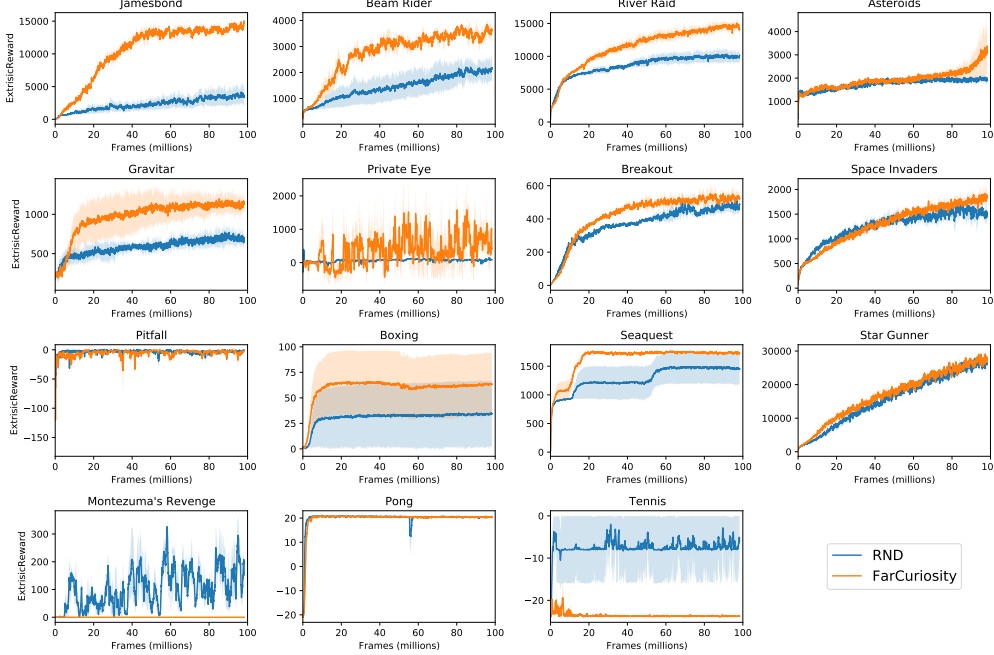

Figure 13: Mean extrinsic reward of RNN-based policies: RND and FarCuriosity with extrinsic reward on fifteen Atari games. The environments are sorted by the heterogeneity measured in Section 4.2: from highest to the lowest

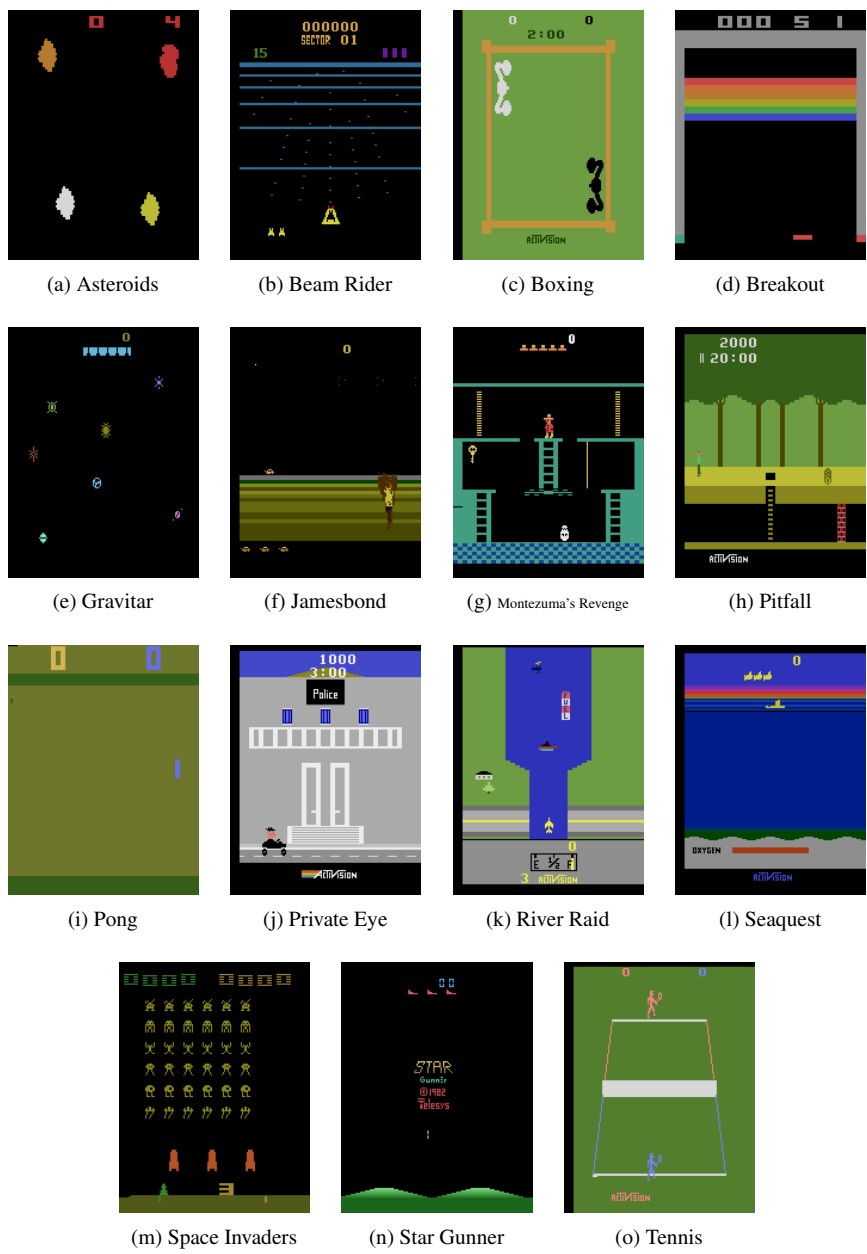

Figure 14: Various Atari environments used for comparing FarCuriosity and RND.

