# OpenReview forum: "Efficient Exploration via Fragmentation and Recall"
_ICLR.cc/2023/Conference — Submitted to ICLR 2023_

### Official Review · Reviewer_m2i7 · 2022-10-24

**Confidence:** 3
**Correctness:** 3
**Technical Novelty And Significance:** 2
**Empirical Novelty And Significance:** 3
**Recommendation:** 5

**Clarity, Quality, Novelty And Reproducibility:**

This work presents an interesting approach to efficient exploration and model-building. However, I believe there are aspects of the presentation that can be improved to make this a more engaging read.
- In figure 2, it is not clear what the observation and its transformed version are. The fact that they are ego-centric top down views with a 120 degrees FOV becomes clear only in Section 4. Also, the shortest-path to the subgoal is a white arrow, and not black as mentioned in the caption.
- It would also be useful to specify how $o_t$ is transformed to $o_t^{\prime}$
- In the FarMap setting, it would be helpful to clearly define terms like cell, visible cell, known cell, occupied cell. Further, is not clear what the action space is.
- It would also be helpful to concretely describe the connectivity graph, i.e., what the nodes and edges correspond to - fragments, fracture points?
- For the results in figure 5, how is memory usage computed?


**Strength And Weaknesses:**

This work is a proof-of-concept for the fragmentation and recall approach to efficient exploration.
The empirical results presented here demonstrate the advantage of this approach. In the spatial exploration task, FarMap achieves comparable map coverage as an improved Frontier-based exploration baseline, but faster and at a substantially lower memory cost. In the RL setting, FarCuriosity obtains higher extrinsic reward than Random Network Distillation (RND) in Atari games with heterogenous environments.

While the initial results presented here are promising, it is not clear how this approach would scale to more complex settings. For instance, in the FarMap setting, the local observation is a top-down view of the environment and has the same structure as the local predictive spatial map.  If the input observation stream is an egocentric view from a 3-D environment, how would it be handled?

It is not clear how the size of the local predictive map $(H, W)$ grows. Also, it is not clear how the observation $o_t$ is transformed to $o_t^{\prime}$. It looks like the size of the current map and the agent's position need to be tracked for this. However, the agent's position might not always be known/cannot be tracked accurately, e.g., when actions are stochastic/noisy.


**Summary Of The Paper:**

This work presents a framework for efficient exploration based on the idea of fragmentation and recall. An agent exploring an environment maintains a local map/model in its short-term memory (STM). At each time step, the agent can compute how well the current observation and the model's prediction of it match. If there is a high mismatch or surprise, a *fragmentation* event occurs, where the agent saves the current local model into a long-term memory (LTM) and initiates a new model. The locations where fragmentation occur are known as fracture points. At such fracture points, the agent is able to *recall* previously stored models, corresponding to overlapping fragments or local models, from its LTM and use these to guide further exploration.

This framework is applied to (i) spatial exploration (FarMap) in 2D grid world type environments , and (ii) curiosity-driven RL (FarCuriosity) in Atari games. In both these settings, the experimental results demonstrate that the fragmentation and recall approach performs better than the respective baselines.

**Summary Of The Review:**

This work presents an interesting approach to efficient exploration based on the idea of fragmentation and recall. While the initial results are promising, I find the current instantiation of the local predictive spatial map limiting. In addition, I think the presentation could be improved to make this a more engaging paper.

---

> ### Author Response · Authors · 2022-11-13
> **Authors' Response to Reviewer m2i7**
>
> We appreciate the reviewer's thoughtful comments. We address each comment below, and we have updated the manuscript to reflect your feedback.
>
>
> **[Scale Up]**
>
> We would like to emphasize that we used 300 procedurally generated environments with 5 different random seeds, which tests FarMap on various shapes of environments. Extensions to stochastic or 3D environments are straightforward:
>
> - **Stochastic Environments**
>
> Frontier-based Exploration (Yamauchi. 1997) was originally devised to address the challenge of noisy sensors (see Section 2 in Yamauchi (1997)). Like the original proposal, we can define known / unknown regions and occupancy based on probability. Although environmental stochasticity can increase surprisal, it does not hugely impact fragmentation since we use the surprisal distribution (i.e., z-score) to decide when to fragment instead of using surprisal directly.
>
> - **3D Environments**
>
> There are many studies on converting egocentric views from 3D environments (e.g., HM3D (Ramakrishnan et al. 2021) and Gibson (Xia et al. 2018)) to a 2D spatial mapping as seen in Chaplot et al. (ICLR. 2020) and Chaplot et al. (NeurIPS. 2020). We can use these methods to convert an egocentric observation from a 3D environment to an observation on a 2D map as a preprocessor. If the agent has access to a 3D sensor such as LiDAR, we can directly build a 3D map instead of converting it to a 2D representation.
>
>
> Ramakrishnan, Santhosh K., et al. "Habitat-matterport 3D dataset (HM3D): 1000 large-scale 3D environments for embodied AI." NeurIPS. 2021.
>
> Xia, Fei, et al. "Gibson env: Real-world perception for embodied agents." CVPR. 2018.
>
> Chaplot, Devendra Singh, et al. "Learning to explore using active neural slam." ICLR. 2020.
>
> Chaplot, Devendra Singh, et al. "Object goal navigation using goal-oriented semantic exploration." NeurIPS. 2020
>
>
>
> **[Clarification of FarMap]**
>
> - **Update local map**
>
> We added a toy illustration of how $o’_t$ is generated and how the predictive map grows in Figure 8. Since the observation is partial and its size is different from the local map, we need to transform it. Given the observation and the current local map, we first rotate the observation based on the agents heading in the local map, and then zero-pad so that the observation can fit in the correct position and has the same size as the local map. If it is impossible since the agent is observing outside of the map (with dimensions $H \times W$), we pad the local map so that it can add the current observation and then transform the observation. In this case, the size of the map is increased. Please refer to Figure 8 in the revised manuscript for a better understanding.
>
> - **Figure 2**
>
> Thank you for pointing this out - we have updated the caption and made it clearer. Black, gray, and dashed arrows denote control flow for planning, fragmentation, and recall, respectively.
>
> - **Cell types**
>
>  A visible cell is a cell that is visible in each observation without occlusion. A known cell is a cell that was visible in the previous or current observation. There are two types of known cells; (1) an occupied cell filled with a wall obstacle, (2) an unoccupied cell without a wall obstacle. An unknown cell is a cell in the local map that hasn’t been observed and therefore has unknown occupancy. A frontier cell is an unknown cell that is adjacent to a known unoccupied cell.
>
> - **Action space**
>
>  In the FarMap environment in our experiment, the action space is going up/down/left/right, and the planner generates a trajectory (a sequence of actions) toward the goal.
>
> - **Connectivity Graph**
>
>  When $\mathbf{M}_t^{cur}$ is stored in LTM, it is connected with an adjacent map fragment that shares the same fracture points in the connectivity graph. Hence, each node is a map fragment and the edge denotes the fracture point. We will clarify it in the revised manuscript.
>
>
> **[How to compute Memory Usage]**
>
> As mentioned by the reviewer RmS4, actual memory usage can vary depending on the implementation. To reduce the difference caused by implementation, we use the size of the local map ($H \times W$) as memory size and define memory usage (%) as the ratio of the local map size to the environment size for calculating the average memory usage across various environments. Note that Frontier, Frontier++, and FarMap require the map and this requirement is asymptotically dominant.

---

> ### Author Response · Authors · 2022-11-17
> **A Letter from the Authors**
>
> Dear Reviewer m2i7,
>
>
> We kindly inform you that the first discussion period will end soon. It would be great if you read our response which can resolve your concerns and comments.
>
> If you have any further comments or questions, we would be happy to address them.
>
> Sincerely,
>
> Authors

---

### Official Review · Reviewer_HdLq · 2022-10-24

**Confidence:** 3
**Correctness:** 3
**Technical Novelty And Significance:** 3
**Empirical Novelty And Significance:** 2
**Recommendation:** 6

**Clarity, Quality, Novelty And Reproducibility:**

The paper is very well- and clearly written, apart from some points I comment below. In addition, the high-quality schematics and Figures help to convey the message and the main intuition behind the algorithms and the use cases addressed. In addition, I find the FarCuriosity application very interesting and useful for addressing the exploration problem in many reinforcement learning settings and problems.

My main point of objection is that the authors do not clearly show (in my opinion) how their work is novel in the FarMap case compared to the previous work of Klukas et al. (2021). In the related work section, in the comparison with Submap-Based SLAM approaches they state: “However, FarMap divides space based on properties of the space (how predictable the space is based on the local map or model), and does so in an online manner using surprisal” but I fail to see how this idea is so fundamentally different compared to the findings of Klukas et al. (2021) that would justify expanding it in half the paper. Maybe it would make more sense that the authors focus on the FarCuriosity angle more in the paper?

Another point regarding reproducibility is the selection of some of the hyper-parameters in the configuration of the algorithms. First of all, what is the threshold \psi mentioned after equation 4? Is this the threshold \rho mentioned later? Also, how the selection (or potential good values) of \epsilon, \rho and number of samples (25) before fragmentation is allowed, would be chosen in a new application? How do you ensure a “good” fragmentation?

A few points about the experiments:
- Please elaborate a bit more why the approach does not perform well on the Tennis game in Atari? Is it a matter of tunning the hyper-parameters that control the fragmentation point selection?
- I would have also expected Go-Explore and Savinov et al. (2019) as baselines in the experiments.
- Why don’t you run experiments on Pitfall and Montezuma’s Revenge as those have also been extensively studied in RND.
- Fig. 11 was actually very nice to see. I wanted to propose such a visualization and was happy to find it in the appendix. You might want to move it to the main paper.

Minor points:
- While first reading the paper I asked myself the following questions / I struggled with the following points:
    - Heterogeneous environments: please formally define when an environment is “heterogeneous” (maybe formally in the paper but also by intuition in the abstract already)
    - What do you mean with “spatial”?  The difference between FarMap and FarCuriosity does not become clear from the beginning
- The first two paragraphs in the introduction are too long. I like motivation over such findings, but it takes too long to get to the point of the paper
- In section 3.1 the way a sub-goal is selected is provided in Eq. 5 in the appendix, but I think it would help text clarity if it is included in this Section.
- Since the classification of environments into homogenous and heterogenous is not a standard term in the RL community, I would suggest that the definition given in the appendix should be moved and discussed in the main text for clarity.
- To add to the previous point, I think the comparison with RND shown in Figure 11, as well as the small discussion on Tennis game could be included in the main text as they provide useful insights on the properties of the algorithm.

Language/Grammer:
- Abstract: On the other hand, … (On the one hand missing before)
- a location a where fragmentation $\rightarrow$ swap a/where
- Subgoal is decided $\rightarrow$ col.
- local map in STM or [the] connectivity graph
- while the latter helps [to] find
- we first find the all frontiers $\rightarrow$ all the frontiers

**Strength And Weaknesses:**

**Strengths:**
- The paper is well-written, and the high-quality figures and system schematics really communicate the intuition behind the paper ideas
- There is extensive and correct use of references, as well as comparison to previous work
- The idea of fragmentation and recall in curiosity-driven Reinforcement Learning is both novel and useful

**Weaknesses:**
- In my opinion, the authors are not clearly demonstrating the necessary novelty step over Klukas et al. (2021), with respect to the FarMap application.
- Also, I find it confusing that two applications (FarMap and FarCuriocity) are described in parallel. Why not focusing on one (maybe the most novel is the FarCuriosity) and provide more evaluation experiments there?
- Even though the authors provide a sensitivity analysis / ablation study for some of the hyper-parameters of the algorithm, it is not clear how these should be tuned (i.e., what would be their boundaries) in a new application / use case.


**Summary Of The Paper:**

The authors propose the utilization of the concept of Fragmentation-and-Recall to solve spatial and reinforcement learning problems. In the former case, which they call FarMap, they address the use case of exploration in sub-map based SLAM and in the latter case, which they call FarCuriosity, the problem of catastrophic forgetting in curiosity-driven RL.

In both cases, the exploring agent focuses on a sub-part of the map or state-space, constructing a local model of exploration (saved in a Short-Term Memory or STM). Once it encounters a novel (also termed “surprising”) new state (called a fragmentation point in the paper), it either stores the sub-map (local map) explored so far to a Long-Term Memory (LTM) or retrieves a sub-map from LTM that is consistent to the new observation. Finally, further exploration is guided through defining sub-goals for the agent either as new points in the LTM or through selecting previous fragmentation points to be reached.

The authors provide experimental results for both cases, using procedurally generated maps for the FarMap application and targeted Atari games in the FarCuriosity case, where they highlight the pros and cons of their method compared to baseline algorithms.


**Summary Of The Review:**

The paper is well written and in a good shape although it may benefit from some re-arrangement and early explanations of crucial concepts. The experimental results on both FarCuriosity and FarMap outperform existing approaches. However, especially for FarMap I am not an expert in the field to judge related work and state of the art to use as baselines. But more importantly, I do not see a major improvement of FarMap ofer Kuskal et al. (2021). On the other side, for FarCuriosity the explanation/discussion (there are a few open questions about the design choices) and the experimental section (I would have expected some specific baselines and environments) could be more thoroughly elaborated.


**Post-Rebuttal**

I would like to thank the authors for their response. I updated my recommendation to 6 as I also share some of the criticism of the other reviewers.

---

> ### Author Response · Authors · 2022-11-13
> **Authors' Response to Reviewer HdLq (1/2)**
>
> We appreciate the reviewer for their thoughtful comments and kind words. We address your comments below and have updated the manuscript to reflect changes.
>
> **[Comparison with Klukas et al. (2021)]**
>
> Klukas et al. (2021) is a conceptual work for neuroscience: it focuses on how surprisal based online fragmentation can make similar fragmentation patterns as seen in recordings of grid cells in the brain. They only give a qualitative example that fracture points can be used as subgoals, and they do so in the context of planning through Figure 5 in the paper. Results in the figure, where the **task is to go from the start to the goal**, require full exploration of the map before planning. Our work is inspired by this neuroscience study, with the goal of implementing the fragmentation and recall concept and applying it to two domains of interest in machine learning; spatial navigation (SLAM) and curiosity-driven reinforcement learning. FarMap employs fragmentation in an exploration setting. We are able to quantitatively show memory and computation gains compared to baselines across 1500 different environments (300 environments x 5 colorizations and starting positions). Moreover, we build a connectivity graph between stored fragments in LTM. Recall can happen if and only if the agent is located at a fracture point, which is different from Klukas et al. (2021), where recall happens in the fragmentation event sampling location from LTM.
>
> **[Why Two Application]**
>
> We propose a general framework, fragmentation-and-recall, that can be applicable to multiple domains. FarMap and FarCuriosity are examples of the domains where the framework is applicable.
>
> **[Hyperparameter]**
>
> Tables 2, 3, and 4 show that FarMap is less sensitive to hyperparameters, \rho, \gamma, \epsilon. The minimum sample size is chosen following the rule of thumb for selecting sufficient samples in statistics, which is usually defined between 25 and 40 to achieve statistical significance.
>
> **[Good Fragmentation in FarMap]**
>
> We compare vanilla FarMap, random fragmentation, and uniform fragmentation methods on top of FarMap to show that our FarMap makes good fragmentations that maintain the exploration performance with low memory and fast wall-clock time. Random and Uniform models only change the fragmentation criteria and other parts (e.g., LTM, subgoal selection, and planning) are the same as FarMap. Table R1 shows that FarMap requires almost half the memory overhead and achieves a 3 times faster runtime when compared with Random (P=P = 0.0005) and Uniform (Interval 2500), which have similar coverage performance.
>
> **Table R1. Comparison of vanilla FarMap and FarMap with random, and uniform fragmentation in Large Environments. The random model decides to fragment with probability P on every time step and the uniform model makes a fragmentation on every Interval step.**
>
> Model | Coverage | Memory | Time
> |:---|---:|---:|---:|
> FarMap | **56.6 (20.8)** | 31.4 (11.8) | 352.5 (110.7)
> |---|---|---|---|
> Random (P = 0.1) |  45.2 (17.9)    |  7.8 (2.0)  | 111.3 (20.8)
> Random (P = 0.05 |  47.5 (18.8)    | 12.1 (3.9)  | 136.5 (27.1)
> Random (P = 0.01) | 49.0 (19.1)    | 24.5 (8.2) |  290.7 (96.3)
> Random (P = 0.005) | 49.1 (19.1)  |  30.5 (11.2) |  378.1 (116.3)
> Random (P = 0.001) | 54.1 (18.4)  |  46.1 (16.2)  | 683.4 (268.8)
> Random (P = 0.0005) |  55.8 (20.1) |    53.9 (18.8) |  914.4 (414.3)
> Random (P = 0.0001) |  57.9 (19.1)  |  61.2 (18.6)  |1181.8 (471.3)
> Random (P = 0.00005) |   58.4 (19.7)   |  63.4 (19.1) | 1269.9 (442.6)
> |---|---|---|---|
> Uniform (Interval 25)|     49.1 (19.3)|      **7.5 (1.8)** |  **110.8 (12.7)**
> Uniform (Interval 50)|     48.8 (19.4)|    12.6 (3.7)|  147.3 (20.9)
> Uniform (Interval 100)|    48.3 (17.6)|     19.3 (5.6)|  216.5 (52.8)
> Uniform (Interval 200)|    48.8 (18.8)|     27.5 (8.6)|  322.2 (112.9)
> Uniform (Interval 500)|    52.2 (19.5)|     38.2 (14.5)|  484.2 (188.4)
> Uniform (Interval 1000)|   53.4 (19.3)|     46.5 (16.7)|  712.3 (403.0)
> Uniform (Interval 2000)|   55.4 (9.2)|     53.8 (16.9)|  848.1 (236.7)
> Uniform (Interval 2500)|   56.0 (19.5)|     56.8 (18.0)|  913.0 (317.8)

---

> ### Author Response · Authors · 2022-11-13
> **Authors' Response to Reviewer HdLq (2/2)**
>
> **[Analysis of Results in Tennis]**
>
> In Tennis (Figure 13. (o)), the agent and opponent play one six-game set of tennis. Like standard tennis, the player and its opponent change their court in every odd game (one gets four points in the game). Therefore, tennis is a very homogeneous environment. However, FarCuriosity generated 44.3 fragments on average in this game (over-fragmentation) and we initialize the curiosity module in STM in the fragmentation event. This might be because the fragmentation criteria is too low for Tennis. We test FarCuriosity on Tennis with a doubled fragmentation threshold and the performance was similar to RND. The average of the final performance in FarCuriosity with a doubled threshold was -1.5 (originally -23.8) while RND was -6.1.
>
> **[Hard Atari Games]**
>
> We additionally test FarCuriosity on standard hard Atari Games like Montezuma’s Revenge, Pitfall, Gravitar, and Private Eye using the same hyperparameters mentioned in Section C.3, and Figures 6 and 12 in the revised manuscript.
>
> FarCuriosity achieves better performance than RND in Gravitar and Private Eye, similar performance in Pitfall, and worse performance in Montezuma’s Revenge. Table R2 summarizes the heterogeneity and the performance improvement in each hard Atari game. Interestingly, the heterogeneity of Montezuma’s Revenge is very low although it is well known for its various rooms. This is because we measure heterogeneity as the average standard deviation of generated trajectories by the trained curiosity module, RND, and it could not explore many rooms. This is directly related to fragmentation rather than the oracle heterogeneity of the game (supposing we can collect all possible observations of the game and calculate their standard deviation). Moreover, it made many fragmentations within one room. This might explain why FarCuriosity is worse than RND in this environment.
>
> **Table R2. Heterogeneity and relative extrinsic reward improvement of Hard Atari Games: Montezuma’s Revenge, Pitfall, Private Eye, Gravitar.**
>
> |Games | Average Standard Deviation (Heterogeneity) | Relative Extrinsic Reward Improvement |
> |:---|---:|---:|
> | Montezuma’s Revenge | 4.8 | -1.0 |
> | Pitfall | 9.4 | 0.8 |
> | Private Eye | 10.9 | 1.5|
> | Gravitar | 12.2 |  0.8|
>
> **[Other baselines]**
>
> Thank you for the suggestion. We would like to emphasize that the goal of FarCuriosity is not to achieve state-of-the-art performance but to address the problem of catastrophic forgetting in prediction-based curiosity methods. Therefore, we only compare with RND. Alternative methods like EpisodicCuriosity (Savinov et al. 2019) did not conduct any experiments in Atari, and Go-Explore allows the agent to restart from any state and heavily exploits human bias, which creates an unfair comparison.
>
> **[Others]**
>
> - **Re: *“What is the threshold \psi mentioned after equation 4?”***
>
> $\psi$ denotes a threshold for recalling the curiosity module in FarCuriosity which is set to 0.99 as mentioned in Section C.3. Since FarCuriosity divides the high dimensional observation space, it requires a threshold for judging whether the agent is currently located in previously explored space by a stored curiosity module.
>
> - **Re: *“What do you mean with “spatial”?  The difference between FarMap and FarCuriosity does not become clear from the beginning”***
>
> Unfortunately, there is a broken text in the question (rectangle). We assume the reviewer's intent is to ask what the difference is between the FarMap (spatial exploration) and FarCuriosity (standard reinforcement learning) tasks. The objective of the FarMap task is to explore a 2D or 3D space efficiently, while the objective of the FarCuriosity task is to get higher extrinsic rewards in potentially non-spatial high-dimensional observation (e.g., the image in Atari, joint coordinates and torques in Mujoco). Note that although some Atari games such as Seaquest and Montezuma’s Revenge look like spatial tasks, the goal of these games is not to explore the space but to achieve higher extrinsic rewards, and the observation is just an 84x84x3 image.

---

> ### Author Response · Authors · 2022-11-17
> **A Letter from the Authors**
>
> Dear Reviewer HdLq,
>
> We kindly inform you that the first discussion period will end soon. It would be great if you read our response which can resolve your concerns and comments.
>
> If you have any further comments or questions, we would be happy to address them.
>
> Sincerely,
>
> Authors

---

### Official Review · Reviewer_FqLm · 2022-10-26

**Confidence:** 4
**Correctness:** 3
**Technical Novelty And Significance:** 3
**Empirical Novelty And Significance:** 2
**Recommendation:** 5

**Clarity, Quality, Novelty And Reproducibility:**

The main idea behind the proposed  Fragmentation-and-Recall framework is space fragmentation by using a surprisal signal. I found the general idea of this work interesting as it allows agents to create local maps that can be reused in the future. As aforementioned, the main weakness of this paper is its clarity a lot of points are missed that could help the reader to understand the proposed framework in depth.
- First of all, the way used to create the local map is vague, i.e. the spatial transformation is not presented at all.
- Also, it is not clear if $M_{t-1,C}^{cur}$ is considered as a vector at Eq.(1).
- Another point that is not clear is the prevention of quite small local maps. How we can avoid the creation of a huge number of local models?
- It is not clear how the agent explores the local map. How a local map is expanded?
- The definition of memory-usage is not clear. How do you compute it?
- Standard deviation or the 90% percentiles should be provided in Fig. 5. Also, the same kind of plots should be provided for the different groups of environment sizes.
- The performance of the FarCuriosity should be examined in hard-exploration environments, such as Montezuma’s revenge.
- How the performance of the FarCuriosity is affected by the number of curiosity modules?
- Source code should be provided. Otherwise, the reproducibility of the results is quite hard or even impossible.

**Strength And Weaknesses:**

**Strengths**

- The proposed Fragmentation-and-Recall framework is general and can be applied in both the spatial exploration (*FarMap*)  and general reinforcement learning exploration  (*FarCuriosity*) settings.
- By using local maps and model fragments we can avoid catastrophic forgetting in learning heterogeneous environments.
- Experiments have been conducted in procedurally-generated spatial environments and on reinforcement learning benchmarks, showing that the proposed method is more efficient regarding memory usage.

**Weaknesses**

- Many of the Fragmentation-and-Recall framework components should be described in more detail.
- The proposed *FarMap* does not outperform Frontier++. Actually, Frontier++ performs much better compared to the  *FarMap* as the map size increases.
- Experiments at hard-exploration environments are missed (i.e. Montezuma’s revenge)



**Summary Of The Paper:**

This work introduces the Fragmentation-and-Recall framework that can be applied for exploration in spatial and general reinforcement learning problems. The idea behind the proposed framework is the fragmentation of the space based on a surprising signal. In this way, agents create local maps (or curiosity modules in the case of RL)  stored in a long-term term and recalled when the agent returns to the state where the fragmentation happened. In this way, local information can be reused by the agent. Authors have applied the framework in the settings of spatial exploration (*FarMap*) and general reinforcement learning exploration (*FarCuriosity*).

**Summary Of The Review:**

In general, the proposed Fragmentation-and-Recall framework is somehow novel and can be applied in both spatial exploration and general reinforcement learning exploration settings. The main limitation of this work is its clarity, as a lot of useful information is missed or doesn't present in depth. Also, the empirical analysis should be considered real maps in the case of spatial tasks and more hard-exploration environments in the RL context.

---

> ### Author Response · Authors · 2022-11-13
> **Authors' Response to Reviewer FqLm (1/2)**
>
> We appreciate the reviewer for the insightful comments. We address your comments below and have updated the manuscript. We will release the code if our paper is accepted.
>
> **[Clarification of Local Map Update]**
>
> We added a toy illustration of how $o’_t$ is generated and how the predictive map grows in Figure 8. Since the observation is partial and its size is different from the local map, we need to transform it. Given the observation and the current local map, we first rotate the observation based on the agents heading in the local map, and then zero-pad so that the observation can fit in the correct position and has the same size as the local map. If it is impossible since the agent is observing outside of the map (with dimensions $H \times W$), we pad the local map so that it can add the current observation and then transform the observation. In this case, the size of the map is increased. Please refer to Figure 8 in the revised manuscript for a better understanding.
>
> **[Memory Usage]**
>
> As mentioned by the reviewer RmS4, actual memory usage can vary depending on the implementation. To reduce the difference caused by implementation, we use the size of the local map ($H \times W$) as memory size and define memory usage (%) as the ratio of the local map size to the environment size for calculating the average memory usage across various environments. Note that Frontier, Frontier++, and FarMap require the map and this requirement is asymptotically dominant.
>
> **[Performance in FarMap compared to Frontier++]**
>
> The goal of FarMap is not to explore unseen spaces faster than the best method, but to reduce memory usage and wall-clock time while maintaining exploration performance. Frontier and Frontier++ are guaranteed to find all frontiers in the environment since they use a global map. On the other hand, FarMap only uses local maps. Therefore, the performance drop is unavoidable. Notably, the performance drop compared to Frontier++ is similar in both medium and large environments (5% and 7%, respectively). We have now run post hoc Dunn test using the holm stepdown method with Bonferonni corrections to check for differences between each model after the Friedman test. The map coverages of Frontier++ and FarMap in Large environments are not statistically significantly different at p = 0.05 (p-value: 0.12).
>
> By contrast, memory usage of FarMap dropped from 73% to 46% of Frontier++ performance as the environment size increased, as did the the wall-clock time, going from 3.4x faster to 5.3x faster for FarMap relative to Frontier++. This means that the memory usage and wall-clock time are improved dramatically while the increment of performance drop is relatively small. In summary, FarMap can be a good option if we have a memory or time constraint.
>
> **[Hard Atari Games]**
>
> We additionally test FarCuriosity on standard hard Atari Games like Montezuma’s Revenge, Pitfall, Gravitar, and Private Eye using the same hyperparameters mentioned in Section C.3, and Figures 6 and 12 in the revised manuscript.
>
> FarCuriosity achieves better performance than RND in Gravitar and Private Eye, similar performance in Pitfall, and worse performance in Montezuma’s Revenge. Table R2 summarizes the heterogeneity and the performance improvement in each hard Atari game. Interestingly, the heterogeneity of Montezuma’s Revenge is very low although it is well known for its various rooms. This is because we measure heterogeneity as the average standard deviation of generated trajectories by the trained curiosity module, RND, and it could not explore many rooms. This is directly related to fragmentation rather than the oracle heterogeneity of the game (supposing we can collect all possible observations of the game and calculate their standard deviation). Moreover, it made many fragmentations within one room. This might explain why FarCuriosity is worse than RND in this environment.
>
> **Table R2. Heterogeneity and relative extrinsic reward improvement of Hard Atari Games: Montezuma’s Revenge, Pitfall, Private Eye, Gravitar.**
>
> |Games | Average Standard Deviation (Heterogeneity) | Relative Extrinsic Reward Improvement |
> |:---|---:|---:|
> | Montezuma’s Revenge | 4.8 | -1.0 |
> | Pitfall | 9.4 | 0.8 |
> | Private Eye | 10.9 | 1.5|
> | Gravitar | 12.2 |  0.8|
>
> **[Performance and the number of curiosity modules in FarCuriosity]**
>
> We added Figure 11 which shows the relation between the relative performance improvement and the average number of fragments on multiple Atari games. In most games, there are at most 50 fragments. The number of fragments reaches the maximum capacity of LTM in Beam Rider. There is no clear tendency between the performance improvement and the number of fragments.

---

> ### Author Response · Authors · 2022-11-13
> **Authors' Response to Reviewer FqLm (2/2)**
>
>  **Re: *“Standard deviation or the 90% percentiles should be provided in Fig. 5. Also, the same kind of plots should be provided for the different groups of environment sizes.”***
>
> In Figure 5, we provided the standard error just as with other plots (Figures 6 and 7 in the original manuscript) for consistency, but since we ran 1500 experiments, it has very small standard errors and is not visible as we mentioned in the main text. We add plots for the different groups of environment sizes and move them to the appendix (Figure 10).
>
> On the other hand, we calculated a 95 \% confidence interval on each group of environments by using bootstrap in Section F of the revised manuscript and the response to RmS4 (Response to Still missing basic experiments (1/3)). In summary, all methods have a huge range of confidence intervals due to the metrics that rely on the size and the complexity of environments. Hence, we compare the relative performance of Frontier++ and FarMap compared to Frontier. Both Frontier++ and FarMap have similar confidence intervals in terms of map coverage while they are different in memory usage and wall-clock time although there are some overlaps.
>
>  ***Re: “Also, it is not clear if $\mathbf{M}_{t-1, C}^\text{cur}$ is considered as a vector at Eq.(1).“***
>
> $\mathbf{M}_{t-1, C}^{cur} \in \mathbb{R}^{H \times W}$ is not a vector but 2D array that contains the confidence of each pixel.
>
>  **Re: *“Another point that is not clear is the prevention of quite small local maps. How we can avoid the creation of a huge number of local models?”***
>
> We collect at least 25 samples before deciding on fragmentation following the statistical rule of thumb of selecting sufficiently many samples. After collecting the surprisal from the current local model (map, curiosity module), the fragmentation happens based on the z-score of the current surprisal. This strategy can implicitly prevent a huge number of fragmentations.
>
> If the environment is complex and leads the agent to generate high surprisal, this high surprisal is already reflected in the surprisal distribution. In other words, much higher surprisal is needed for fragmentation. On the other hand, if the environment is mixed with complex regions and simple regions, it is possible to make a lot of local models but this is the correct fragmentation.

---

> ### Author Response · Authors · 2022-11-17
> **A Letter from the Authors**
>
> Dear Reviewer FqLm,
>
> We kindly inform you that the first discussion period will end soon. It would be great if you read our response which can resolve your concerns and comments.
>
> If you have any further comments or questions, we would be happy to address them.
>
> Sincerely,
>
> Authors

---

### Official Review · Reviewer_RmS4 · 2022-11-03

**Confidence:** 4
**Correctness:** 2
**Technical Novelty And Significance:** 2
**Empirical Novelty And Significance:** 1
**Recommendation:** 5

**Clarity, Quality, Novelty And Reproducibility:**

In general the paper clarity is ok, but should be better. Mainly this could be achieved by being more precise in many minor statements throughout the paper. See some of the notes below.

This work has similar feeling to work on growing basis functions in RL. See Samejima and Omori (1999), Whiteson et al. (2007), Ure et al. (2012), and Mitchley (2015) for examples of this work. Specific these works grow new basis functions to expand the function approximator's ability to better predict the value function or transition dynamics based on having large prediction errors. It is very similar to the fragmentation idea presented in this work.



Misc notes:
At the start of section 3, there are many repeated points. It would be beneficial to be more direct and introduce the method with a mathematical description and not rely solely on words.

Page 5: What is a frontier cell? It was not yet defined and not defined until much later.

Page 6: What does it mean for a set of states to be diverse?

In Figure 6, use different units to make the numbers more readable.

Page 8: "we present the empirical results that verify the problem of catastrophic forgetting of intrinsic rewards in reinforcement learning." The problem of catastrophic forgetting is not a problem in reinforcement learning per se. It would be more accurate to say it is a problem with function approximations used in RL. Tabular methods do not have a catastrophic forgetting problem.

Mitchley, M. R. (2015). Adaptive Value Function Approximation in Reinforcement Learning using Wavelets (Doctoral dissertation, University of the Witwatersrand, Faculty of Science, School of Computational and Applied Mathematics).

Samejima, K., & Omori, T. (1999). Adaptive internal state space construction method for reinforcement learning of a real-world agent. Neural Networks, 12(7-8), 1143-1155.

Ure, N. K., Geramifard, A., Chowdhary, G., & How, J. P. (2012, September). Adaptive planning for Markov decision processes with uncertain transition models via incremental feature dependency discovery. In Joint European conference on machine learning and knowledge discovery in databases (pp. 99-115). Springer, Berlin, Heidelberg.

Whiteson, S., Taylor, M. E., Stone, P. (2007). Adaptive tile coding for value function approximation (No. AI-TR-07-339).

**Strength And Weaknesses:**

The idea forgoes the current trend of having monolithic forms of function approximation and tries to address an important problem with many algorithms. However, some key issues need to be addressed before the paper is ready for publication. The primary issue is that the experiments do not substantiate the claims or provide meaningful insights into the algorithms. I will detail these below.

For the FarMap and FarCuriosity experiments, each method is run on multiple environments to show that the new methods outperform the existing ones. There are several issues with the experimental setup. The first is that only one selection of hyperparameters is considered. As has been pointed out several times (Henderson et al. 2018, Jordan et al. 2020, ml ones), this leads to arbitrary comparisons in claiming that one method outperforms another. So these experiments cannot say one method is better than another outside of these specific hyperparameter choices. This experiment is of little value because the hyperparameters need to be tuned for different environments.

Another issue in the experiments is the comparison of memory usage. The % of memory used is not an informative metric because it depends on the system's memory (more of a minor issue). It would be better to show the actual memory usage. Also, the amount of memory used by each method highly depends on the implementation. There can be numerous code optimizations to reduce the memory of each method. It is worth considering a metric independent of the actual implementation to provide a fair comparison. Showing the actual memory usage of a specific implementation in addition to the other metric would provide a realistic context of what one could expect. Similar to the other experiments, the results here also depend on the hyperparameters, and their effect must be considered.

To address the hyperparameter issues, one could consider experiments to answer other questions that would be more helpful than a performance comparison. A few possible questions: 1) how does changing the number steps before fragmentation impact the number of local maps, and 2) how sensitive is the behavior of the algorithm to epsilon? These are important questions if you want someone to be able to use the algorithm successfully.

Figure 5: Standard error is not an accurate representation of confidence. Also, how was it computed because the number of trials is not 1500, as stated on page 7? The run of the algorithm on different problems is not an i.i.d sample, so you cannot lump them all together to compute the variance.

In Table 1, if you want to compare the performance of each algorithm, you should use confidence intervals to identify if one method is better than another, not the empirical measure. Based on the large standard deviations, it is unclear whether one method is better than another. Remember to correct for multiple comparisons amongst the algorithms when making these comparisons.

Page 8: "On the other hand, RND generates a higher intrinsic reward (higher prediction error) as training progresses, implying that this method suffers from catastrophic forgetting"
The experiments are not a valid test of catastrophic forgetting taking place. If you want to measure catastrophic forgetting, then measure it with an experiment. This statement is simply a guess at what is going on. However, this is a great hypothesis that should be tested, i.e., does RND accurately predict the intrinsic reward signal at previously seen states later in training? Then follow it up with a question based on your approach. Does splitting the function approximator at points of high prediction error provide longer-term memory of the intrinsic reward signal? Performance-based experiments can only be used to test the differences in performance, not underlying factors like catastrophic forgetting.

A few questions:
Page 5: is the backward direction backward in time? Because the agent could be facing any direction. Defining the method with math would make things more precise and avoid confusion.

In what sort of problems is the FarMap applicable? The prediction of the observation is not a true model of what occurs, e.g., the agent could move left or right, and the local map could not predict the outcome. Additionally, would it not induce many fragmentations if the colors change in the map with a high frequency?

What feature extractor was used in the RL experiments?

References:
Henderson, P., Islam, R., Bachman, P., Pineau, J., Precup, D., and Meger, D. Deep reinforcement learning that matters. In Proceedings of the Thirty-Second AAAI Conference on Artificial Intelligence, (AAAI-18), pp. 3207–3214, 2018.

Jordan, S., Chandak, Y., Cohen, D., Zhang, M., & Thomas, P. (2020, November). Evaluating the performance of reinforcement learning algorithms. In International Conference on Machine Learning (pp. 4962-4973). PMLR.

Lucic, M., Kurach, K., Michalski, M., Gelly, S., and Bousquet, O. Are gans created equal? A large-scale study. In Advances in Neural Information Processing Systems 31., pp. 698–707, 2018.

Melis, G., Dyer, C., and Blunsom, P. On the state of the art of evaluation in neural language models. In 6th International Conference on Learning Representations, ICLR. OpenReview.net, 2018.


**Summary Of The Paper:**

This paper proposes a function approximation framework for mapping and exploration in RL. The function approximation scheme uses multiple local models that predict the next observation, and new local models get created when a prediction error is above a user-defined threshold. The local model used at each step is determined by the similarity of the current observation to the observation stored in memory for each local model. The mapping procedure also creates a graph of the local models to allow for exploration across local regions. In the reinforcement learning setting, the local models' intended benefit is overcoming catastrophic forgetting, which occurs when using a single neural network to predict intrinsic reward signals.

**Summary Of The Review:**

The topic of the paper is worth further study, but due to poor experimentation this paper is not yet ready for publication.

--update--
After discussing with the authors and seeing changes to the paper, it has improved, but still lacks experiments that demonstrate the method is behaving as intended. I updated my score to a 5 marginally below the threshold.

---

> ### Author Response · Authors · 2022-11-13
> **Authors' Response to Reviewer RmS4 (1/2)**
>
> We appreciate the reviewer for the thoughtful comments. We address your comments below and updated the manuscript.
>
> **[Hyperparameter]**
>
> We conducted a sensitivity analysis of $\rho$, $\gamma$, and $\epsilon$ for the FarMap in Tables 2, 3, and 4 in Appendix, respectively. FarMap is not sensitive to these hyperparameters.
>
> **[Memory Usage]**
>
> As mentioned by the reviewer RmS4, actual memory usage can vary depending on the implementation. To reduce the difference caused by implementation, we use the size of the local map ($H \times W$) as memory size and define memory usage (%) as the ratio of the local map size to the environment size for calculating the average memory usage across various environments. Note that Frontier, Frontier++, and FarMap require the map and this requirement is asymptotically dominant.
>
> **[Confidence Interval in Figure 5]**
>
> We used standard error which is identical to the 68% of confidence interval in Figures 5, 6, 7, and 10 (Figures 10, 5, and 12 in the revised manuscript).
>
> **[Statistical Analysis of FarMap Experiments]**
>
> We conducted the Friedman Test on Frontier, Frontier++, and FarMap to see whether all achieve the same performance of not on for each group of environments.
>
> **Table R1.  The results of the Friedman test. the data is paired by each environment.**
> Environment Group |  Coverage - Statistics | Coverage - p-value | Memory - Statistics | Memory - p-value | Runtime -  Statistics | Runtime  - p-value
> |:---|---:|---:|---:|---:|---:|---:|
> All | 923.8 | 2.5e-201 | 888.9 | 9.6e-194 | 742.4 | 6.2e-162
> Small | 774.4 | 6.9e-169 |  294.3 | 1.2e-64 | 375.0 | 3.8e-82
> Medium | 274.6 | 2.4e-60 | 379.6 | 3.7e-83 | 498.5 | 5.6e-109
> Large | 170.2 | 1.1e-37 | 211.8 | 9.9e-47 | 229.1 | 1.8e-50
>
> Then, we ran the posthoc Dunn test using the holm stepdown method with Bonferonni corrections to check for differences between each model in terms of map coverage, memory usage, and wall-clock time. The subject (pair) is each environment and random seed. Only the difference in map coverages of Frontier++ and FarMap in Large environments is not statistically significant at p = 0.05 (p-value: 0.12) and other differences are statically significant.
>
> **[Catastrophic Forgetting]**
>
> In Figure 8 (Figure 7 in the revised manuscript), we show the intrinsic rewards for the same starting observation generated by RND and FarCuriosity during training (the agent is trained in standard episodes and we only log the intrinsic reward from the starting observation). RND’s intrinsic reward keeps increasing as training goes on. Considering the fact that intrinsic reward in RND is defined as the prediction error to a fixed target network from the training predictor network, increasing prediction error means that RND forgets the information of the observation although the observation occurs across the training.
> On the other hand, intrinsic reward from FarCuriosity decreases after showing an increase in the early stage of training.
>
> **[Comparison with growing basis function in RL literature]**
>
> The goal of growing basis functions in reinforcement learning is to estimate the value function or transition dynamics by summing over multiple basis functions. These functions are created by adaptive basis division (Samejima and Omori. 1999), multiscale adaptive wavelet basis (Mitchy. 2015) adaptive tile coding (Whiteson et al. 2007). The creation of a new basis function is based on TD error or Bellman error.  Ure et al. (2015) estimate state-dependent uncertainties in Markov Decision Process (MDP).
>
> Our fragmentation-and-recall framework is inspired by chunking in psychology and grid cells in neuroscience. Both growing basis functions in RL and our FarCuriosity generate multiple local models. The growing basis is purposed for approximating a function (mostly value functions), while our fragmented models, FarMap and FarCuriosity, are not aimed at approximating a function, but rather a memory model that is agnostic to basis functions. Besides, FarCuriosity only uses one model fragment each time and recalls stored model fragments in LTM based on the input observation while growing basis functions in RL employ all basis functions at the same time.  Furthermore, FarCuriosity uses a standard policy optimization method (PPO) and focuses on the intrinsic reward function (curiosity module). In other words, FarCuriosity does not aim to make a better or more efficient value function approximator.
>
> A better basis function may be complementary to our method and make it more memory efficient, while the basis function is not our focus. That being said, growing basis and our method share a similar abstract idea of partitioning the inputs and processing them using localized components (i.e., local fragmentation or basis functions).

---

> ### Author Response · Authors · 2022-11-14
> **Authors' Response to Reviewer RmS4 (2/2)**
>
>
> **[Application of FarMap]**
>
> Although we used 300 procedurally generated environments with 5 different random seeds to show FarMap works well in various shapes of environments, it can be applicable to more difficult environments such as noisy or 3D environments:
>
> - **Noisy or Stochastic Environments**
>
> Frontier-based Exploration (Yamauchi. 1997) was originally devised to tackle noisy sensors (see Section 2 in Yamauchi (1997)). Like the original proposal, we can define  known / unknown regions and occupancy based on its probability. Moreover, although stochastic environments increase the surprisal, it does not hugely impact the fragmentation since we use the distribution of the surprisal (i.e., z-score) instead of directly using it.
>
> - **3D or real environments**
>
> There are many studies to convert egocentric views from 3D environments such as HM3D (Ramakrishnan et al. 2021) and Gibson (Xia et al. 2018) to 2D spatial mappings such as Chaplot et al. (ICLR. 2020) and Chaplot et al. (NeurIPS. 2020). We can use these algorithms to convert egocentric observations in 3D environments to 2D maps as a preprocessor. If the agent can access to the 3D sensor such as LiDAR, we can directly build a 3D map instead of 2D in the same way.
>
> Ramakrishnan, Santhosh K., et al. "Habitat-matterport 3D dataset (HM3D): 1000 large-scale 3D environments for embodied AI." NeurIPS. 2021.
>
> Xia, Fei, et al. "Gibson env: Real-world perception for embodied agents." CVPR. 2018.
>
> Chaplot, Devendra Singh, et al. "Learning to explore using active neural slam." ICLR. 2020.
>
> Chaplot, Devendra Singh, et al. "Object goal navigation using goal-oriented semantic exploration." NeurIPS. 2020.
>
> **[Others]**
>
> - **Re: "*Page 5: is the backward direction backward in time?”***
>
> The backward direction is the spatial direction where the agent came from (spatially behind the agent). We revised the manuscript more clearly.
>
> - **Re: “*Additionally, would it not induce many fragmentations if the colors change in the map with a high frequency?”***
>
> If the wall colors are changing frequently (stochastic environment), the surprisal distribution is shifted to having a higher mean. However, since we do not use the surprisal directly but use the distribution of the surprisal (i.e., z-score), it does not induce many fragmentations.
>
> - **Re: *“Page 5: What is a frontier cell?”***
>
> As mentioned in the second sentence in Section 2 and the second paragraph of Section 3.1. Subgoal, the frontier cell is a cell between observed (known) and unobserved (unknown) cells in the map. To be specific, a known cell is a cell that was visible in the previous or current observation. There are two types of known cells; (1) an occupied cell filled with a wall obstacle, (2) an unoccupied cell without a wall obstacle. An unknown cell is a cell in the local map that hasn’t been observed and therefore has unknown occupancy. A frontier cell is an unknown cell that is adjacent to the known unoccupied cell.
>
> - **Re: *“What feature extractor was used in the RL experiments?”***
>
> We use RND’s fixed network as the feature extractor.
>
> - **Re: “*Page 6: What does it mean for a set of states to be diverse? “***
>
> “a diverse set of states” in Section 3.2 means that the state space has high variance which could be the result of having multiple rooms and stages having different backgrounds.
>
> - **Re: “*In Figure 6, use different units to make the numbers more readable.”***
>
> Thank you for the suggestion. We update Figure 6 (Figure 5 in the revised version).
>
> - **Re: Page 8: "we present the empirical results that verify the problem of catastrophic forgetting of intrinsic rewards in reinforcement learning." The problem of catastrophic forgetting is not a problem in reinforcement learning per se. It would be more accurate to say it is a problem with function approximations used in RL. Tabular methods do not have a catastrophic forgetting problem.**
>
> Thank you for pointing that out. We will clarify that we are only focusing on Deep Reinforcement Learning.

---

> > ### Comment · Reviewer_RmS4 · 2022-11-14
> > **Still missing basic experiments**
> >
> > I want to thank the authors for their detailed responses and revision efforts.
> >
> > First, I want to discuss using standard error as a confidence interval on the mean. The standard error only produces a valid 68% confidence interval for normally distributed random variables; the random variables it is applied to in the paper are not normally distributed. Assuming the sample mean is normally distributed, one can use the t-distribution to construct confidence intervals around the sample mean. Another point is that the 1500 samples are not the degrees of freedom because they are not all i.i.d. If the average performance from each seed is used, then there would be 300 samples that could be used to compute the variance and confidence intervals. Additionally, 68% is a low confidence level; 95% would be more appropriate.
> >
> > In your response, you listed p-values in a table, which is known to be misleading. It is better to report confidence intervals for each entry; if they do not overlap, there is a statistically significant difference.
> >
> > My central issue with the experiments in the paper is that they do not demonstrate that catastrophic forgetting is happening and that the new method does not have catastrophic forgetting. Any claims based on the performance of one algorithm versus another are merely a guess that the method addresses the stated problem. There should be experiments that measure and compare the forgetting of each method.
> >
> > The discussion on the relation to growing basis functions was not to suggest that your method is covered by those methods, only that there are strong similarities. In your response, you say: "FarMap and FarCuriosity, are not aimed at approximating a function," but that is precisely what these methods are doing. The map is a function that predicts the next observation, and the goal is to approximate that function more efficiently. In Far Curiosity, the goal is explicitly function approximation, it just isn't value function approximation, but that is an irrelevant detail to the concepts employed.

---

> > > ### Author Response · Authors · 2022-11-17
> > > **Response to Still missing basic experiments (1/3)**
> > >
> > > We appreciate the response from the reviewer.
> > >
> > > **Re: ***"First, I want to discuss using standard error as a confidence interval on the mean. The standard error only produces a valid 68% confidence interval for normally distributed random variables; the random variables it is applied to in the paper are not normally distributed."*****
> > >
> > > Thank you for pointing out, since the distribution is not normal, the standard error is not the same as the 68% confidence interval.
> > >
> > > **Re: ***"Assuming the sample mean is normally distributed, one can use the t-distribution to construct confidence intervals around the sample mean***”**
> > >
> > > Since the distribution is non-normal we also cannot use the t-distribution. Rather, we conducted Bootstrapping with one million samples with replacement from the experimental results.
> > >
> > > Tables R2, R3, and R4 show that confidence intervals overlap. This overlap is because relative performance depends on the environment and the environments are sampled from a distribution with very different sizes and shapes (please refer to the attached zip file to look at the environments).
> > >
> > > Therefore, as shown in the previous response, we conducted **Friedman Test (Friedman. 1937) followed by Dunn Test (Dunn. 1961)** to statistically compare the methods in each environment. The result presented that all methods and metrics are statistically different, except for map coverage of Frontier++ and FarMap in Large environments, which are similar, as we would like to claim.
> > >
> > > On the other hand, we present the normalized performance of Frontier++ and FarMap to Frontier in each environment in Tables R4, R5, and R6. Moreover, we added a new plot for the confidence interval of the normalized performance. We sorted the environment size from the smallest to the largest and grouped them by 50 environments. Then, we run bootstrap for each group with one million samples to calculate a 95% confidence interval. The tables and the figure show that the map coverage performances of Frontier++ and FarMap are similar. Memory usage and wall-clock time are more different as the environment size gets bigger although there are overlaps in terms of a 95% confidence interval.
> > >
> > > **Table R2. Average and 95% Confidence of Map Coverage on Various Environment Groups from Bootstrap (N=1M).**
> > > Method  |  Small | Medium | Large
> > > |:---|---:|---:|---:|
> > > Frontier |97.2 (76.0, 100.0) | 76.3 (15.6, 99.8) | 41.4 (6.1, 84.3)
> > > Frontier++ | 98.8 (97.5, 100.0)| **91.4 (15.2, 99.8)** | **60.8 (6.1, 98.7)**
> > > FarMap | **99.0 (96.3, 100.0)** |  86.4 (15.6, 100.0) | 56.6 (6.1, 97.7)
> > >
> > > **Table R3. Average and 95% Confidence of Memory Usage on Various Environment Groups from Bootstrap (N=1M).**
> > > Method |  Small| Medium | Large
> > > |:---|---:|---:|---:|
> > > Frontier  | 80.4 (61.8, 88.7) | 73.3 (13.0, 92.3) | 44.4 (3.8, 84.3)
> > > Frontier++ | 81.6 (71.1, 88.7) | 85.9 (12.5, 93.3) | 68.1 (3.8, 94.2)
> > > FarMap | **79.1 (61.4, 88.0**) | **62.9 (12.5, 90.2)** | **31.4 (3.8, 54.3)**
> > >
> > > **Table R4. Average and 95% Confidence of Wall-Clock Time on Various Environment Groups from Bootstrap (N=1M).**
> > > Method  |  Small | Medium | Large
> > > |:---|---:|---:| ---:|
> > > Frontier |   360.5 (154, 773) | 871.9 (290, 2020) | 1261.0 (217, 3189)
> > > Frontier++ | 341.1 (118, 852) | 1099.5 (365, 2339) | 1874.4 (421, 3785)
> > > FarMap |  **278.2 (139, 538)** | **321.4 (191, 528)** | **352.5 (202, 633)**
> > >
> > > **Table R5.  Average and 95% Confidence of Map Coverage on Various Environment Groups from Bootstrap (N_sample = 1M). The result is normalized by the result of Frontier in each environment.**
> > > Method   |  Small | Medium | Large
> > > |:---|---:|---:| ---:|
> > > Frontier++ | 1.02 (0.99, 1.20) |  **1.26 (0.99, 2.30)** | **1.65 (1.00, 3.30)**
> > > FarMap | **1.03 (0.99, 1.20)** | 1.20 (0.79, 2.20) | 1.56 (0.79, 3.20)
> > >
> > > **Table R6. Average and 95% Confidence of Memory Usage on Various Environment Groups from Bootstrap (N_sample = 1M). The result is normalized by the result of Frontier in each environment.**
> > > Method  |  Small | Medium | Large
> > > |:---|---:|---:|---:|
> > > Frontier++ | 1.03 (1.00, 1.20) | 1.24 (0.97, 2.20) | 1.78 (1.00, 3.80)
> > > FarMap| **1.00 (0.72, 1.20)** | **0.90 (0.42, 1.60)** | **0.84 (0.37, 1.70)**
> > >
> > > **Table R7.  Average and 95% Confidence of Wall-Clock Time on Various Environment Groups from Bootstrap (N_sample = 1M). The result is normalized by the result of Frontier in each environment.**
> > >
> > > Method  |  Small | Medium | Large
> > > |:---|---:|---:| ---:|
> > > Frontier++ | 0.95 (0.56, 1.60) | 1.36  (0.68, 2.70)  | 1.75 (0.80, 4.80)
> > > FarMap |  **0.85 (0.41, 1.70)** | **0.46 (0.16, 1.30)** | **0.37 (0.13, 1.30)**
> > >
> > > Friedman, Milton. "The use of ranks to avoid the assumption of normality implicit in the analysis of variance." *Journal of the american statistical association* 32.200 (1937): 675-701.
> > >
> > > Dunn, Olive Jean. "Multiple comparisons among means." *Journal of the American statistical association*  56.293 (1961): 52-64.

---

> > > ### Author Response · Authors · 2022-11-17
> > > **Response to Still missing basic experiments (2/3)**
> > >
> > > **Re:** ***“In your response, you listed p-values in a table, which is known to be misleading. It is better to report confidence intervals for each entry; if they do not overlap, there is a statistically significant difference.”***
> > >
> > > The **Friedman Test (Friedman. 1937)** is designed to test the null hypothesis that the means of groups with paired samples across multiple conditions are all the same. We chose this method since we conducted different experiments using the same group of environments, hence the data are paired. As shown in Table R1, all p-values are almost zero which means that on every environment set (All, Small, Medium, and Large), there are at least two methods that have means whose difference is statistically significant
> > >
> > > To find which pairs of methods have means whose difference is statistically significant, we conduct **Dunn test (Dunn. 1961)** as a Posthoc analysis. Map coverage of Frontier++ and FarMap is the only pair whose p-value is greater than 0.05, which means that their difference is not statistically significant.
> > >
> > > **Re:** “***Another point is that the 1500 samples are not the degrees of freedom because they are not all i.i.d. If the average performance from each seed is used, then there would be 300 samples that could be used to compute the variance and confidence intervals. Additionally, 68% is a low confidence level; 95% would be more appropriate.”***
> > >
> > > Depending on the random seed, the color of the map and the starting position are also changed even if the entire structure is the same, as mentioned in Section 4.1. We will clearly differentiate the terms map and environment in the revised manuscript.
> > >
> > > **Re:** “***The discussion on the relation to growing basis functions was not to suggest that your method is covered by those methods, only that there are strong similarities.***
> > > ***In your response, you say: "FarMap and FarCuriosity, are not aimed at approximating a function," but that is precisely what these methods are doing.***
> > > ***The map is a function that predicts the next observation, and the goal is to approximate that function more efficiently.***
> > > ***In Far Curiosity, the goal is explicitly function approximation,***
> > > ***it just isn't value function approximation,***
> > > ***but that is an irrelevant detail to the concepts employed.***"
> > >
> > > We agree that the fragmentation picture presented in FarMap/FarCuriosity is similar in some very general/high-level sense with growing basis functions and Chinese Restaurant Process: they all grow the model expressivity over time. However, we believe that expressivity is growing in a different way in growing basis function/CRP type models versus in the fragmentation approach. In the former, there is a global world model that is being enriched with more basis functions with time and experience. This single global world model is used everywhere in the state-action space: the basis functions do not necessarily partition the state-action space. In the fragmentation approach of FarMap/FarCuriosity, there is an increasing number of local models with time and experience, but these local models partition the state-action space: only one local model is used in each partition of the state-action space.
> > >
> > > (To kindly reiterate our fragmentation-and-recall framework inspired by neuroscience and psychology. Our framework consists of two memory; short-term memory (STM) and long-term memory (LTM). STM has a local model fragment that the agent is currently using and LTM stores unused fragments (with information on the fracture point). If the agent is located in the fracture point (in FarMap) or has a similar observation as in any fracture point, the corresponding model fragment in LTM is recalled. Consequently, in every time step, only one model is used.)

---

> > > ### Author Response · Authors · 2022-11-17
> > > **Response to Still missing basic experiments (3/3)**
> > >
> > > **Re: *“My central issue with the experiments in the paper is that they do not demonstrate that catastrophic forgetting is happening and that the new method does not have catastrophic forgetting. Any claims based on the performance of one algorithm versus another are merely a guess that the method addresses the stated problem. There should be experiments that measure and compare the forgetting of each method.”***
> > >
> > > We agree with the reviewer that it is important to directly demonstrate our claim of catastrophic forgetting in RND versus FarCuriosity, and we apologize that in both the original submission and response we were just not sufficiently clear about what we have done to show this — we believe that Figure 7 does directly demonstrate what the reviewer is asking, but did not communicate it properly. Below, we first discuss catastrophic forgetting in supervised versus reinforcement learning and then clarify what Figure 7 is showing and why we believe it is a reasonable measure of catastrophic forgetting.
> > >
> > > The standard way to show catastrophic forgetting is to train sequentially on a set of tasks, and then go back and test accuracy on earlier tasks (Douillard et al. 2020; Kemker et al. 2018; Lopez-Paz and Ranzato 2017; Prabhu et al. 2020; Wołczyk et al. 2021). Catastrophic forgetting is considered as “negative backward transfer” (i.e. performance drops on earlier tasks while training subsequent tasks) (Lopez-Paz and Ranzato 2017). Because in supervised learning, data are presented IID within each task, catastrophic forgetting only happens between tasks.
> > >
> > > By contrast, in our setup and in RL more generally, catastrophic forgetting can happen even within a single task, because the data distribution seen by the agent depends on the agent’s actions: in other words, within a task data are not seen IID, and returning to the same external state after extensive training within the same environment can result in catastrophic forgetting.
> > >
> > > Therefore, our measure of catastrophic forgetting for both RND and FarCuriosity is to index, over the course of learning in a given environment, all the times the agent returns to the same initial state, and then quantify the prediction error of the agent (measured as MSE/intrinsic reward) in that state. This is what is plotted in Figure 7, though we believe we should clarify the axes of the plot and the caption and text related to the figure to clarify that this is what we did. In other words, Figure 7 shows that when the RND agent returns to the same state in the environment after further learning in the environment, its prediction error (measured by intrinsic reward) grows, while the error of the FarCuriosity agent at that state remains the same after a short initial increase, regardless of how much subsequent learning the agent performs elsewhere in the environment.
> > >
> > > We hope this helps clarify and shows that are indeed quantifying what the reviewer suggested.
> > >
> > >
> > > Douillard, Arthur, et al. "Podnet: Pooled outputs distillation for small-tasks incremental learning." *ECCV.* 2020.
> > >
> > > Kemker, Ronald, et al. "Measuring catastrophic forgetting in neural networks." *AAAI*. 2018.
> > >
> > > Kirkpatrick, James, et al. "Overcoming catastrophic forgetting in neural networks." *PNAS.* 114.13 (2017): 3521-3526.
> > >
> > > Lopez-Paz, David, and Marc'Aurelio Ranzato. "Gradient episodic memory for continual learning." *NeurIPS.* 2017
> > >
> > > Prabhu, Ameya, Philip HS Torr, and Puneet K. Dokania. "Gdumb: A simple approach that questions our progress in continual learning." *ECCV.* 2020.
> > >
> > > Wołczyk, Maciej, et al. "Continual world: A robotic benchmark for continual reinforcement learning." *NeurIPS.* 2021.

---

> > > > ### Comment · Reviewer_RmS4 · 2022-11-22
> > > > **Good changes**
> > > >
> > > > Ah yes Figure 7 does show what I was asking to see! This was not clear (missing) initially. Checking at the start state is good, but not a thorough experiment. I recommend saving several states including one that would not be seen early in training and check for their reward prediction throughout training.
> > > >
> > > > In Figure 7 it looks like FarCuriosity is still increasing in value although slower than RND. I would plot the change in value to see if it is actually plateaued or not. Furthermore, shouldn't the intrinsic reward go to zero for the start state since it is visited every episode? Something seems off there.
> > > >
> > > > Conducting more experiments that develop understanding of how the method works would make this paper great.

---

> > > > > ### Author Response · Authors · 2022-11-22
> > > > > **Response to Comments regarding Catastrophic Forgetting**
> > > > >
> > > > > We appreciate the reviewer for the suggestions and comments.
> > > > >
> > > > > **Re: *“I recommend saving several states including one that would not be seen early in training and check for their reward prediction throughout training.”***
> > > > >
> > > > > Thank you for the suggestion. We will conduct experiments and describe the results if we can get them within the discussion period.
> > > > >
> > > > > We also recorded the average of the intrinsic reward of the first 128 frames although we note that the frames vary after the first frame (i.e. there are multiple possible values for the second frame). The reward curves have a similar shape to that of the first frame.
> > > > >
> > > > > **Re: *“In Figure 7 it looks like FarCuriosity is still increasing in value although slower than RND. I would plot the change in value to see if it is actually plateaued or not.”***
> > > > >
> > > > > Since we cannot revise the paper in discussion period 2, we performed a linear regression fit on the curve from 25 million frames to 100 million frames instead. Table R5 shows the linear regression for the intrinsic reward of RND and FarCuriosity in Figure 7. As the reviewer pointed out, FarCuriosity’s coefficient is not zero. However, it is 3.4 times smaller than RND’s. This means that although our curiosity module in FarCuriosity covers a much smaller observation space than vanilla RND, catastrophic forgetting slightly happened in FarCuriosity, which might be due to the nature of reinforcement learning (states are not iid in an RL setting).
> > > > >
> > > > > However, we would like to emphasize that FarCuriosity’s degree of forgetting (increment of prediction error) is much smaller than RND’s.
> > > > >
> > > > > We will revise the corresponding part in Section 4.2 (`The error of the FarCuriosity agent at the observation remains the same after a short initial increase,`)
> > > > >
> > > > > **Table R5 Linear Regression of Figure 7 from 25 million frames to 100 million frames.**
> > > > >
> > > > > Method | Coefficient |  Intercepts | $R^2$
> > > > > |:---|---:|---:|---:|
> > > > > RND  |  6.46e-3 | 0.66 | 0.78
> > > > > FarCuriosity | 1.90e-3 | 0.84 | 0.41
> > > > >
> > > > > **Re: “*shouldn't the intrinsic reward go to zero for the start state since it is visited every episode?”***
> > > > >
> > > > > That is a good point! In our opinion, people do not worry about catastrophic forgetting in Atari or other reinforcement learning benchmarks since the episode is usually reset. We also had thought that before. However, it can happen as the start state is less frequently appeared in training. At the beginning of training, the agent dies quickly or explores near the start state, so it is more likely to be in the start state. However, as the agent trains, its episodes become longer and the visited state is more diverse, so it visits the start state less frequently (i.e., several mini-batch does not have the start state). This leads to catastrophic forgetting of the start state.
> > > > >
> > > > > That is why we believe that our work can also contribute to the reinforcement learning community by introducing catastrophic forgetting issues in curiosity-driven reinforcement learning and tackling this problem first.

---

> > > > > ### Author Response · Authors · 2022-12-03
> > > > > **Linear Coefficients for intrinsic rewards of other observations**
> > > > >
> > > > > **Re: *“I recommend saving several states including one that would not be seen early in training and check for their reward prediction throughout training.”***
> > > > >
> > > > > For each run (3 RND, 3 FarCuriosity), we collected intrinsic rewards for observations that occurred at least 25 times after 25 million frames in Jamesbond. We note that these observations were not observed before the 25 million frames. We then ran linear regression between running progress and the intrinsic reward of each observation to get a corresponding linear coefficient, and we present their mean confidence intervals (68% and 95%) in Table R6. Although there is an overlap between FarCuriosity and RND in the 95% confidence interval, the mean is always positive in RND and consistently negative in FarCuriosity.
> > > > >
> > > > > **Table R6. Mean and 68%, 95% confidence interval of linear coefficients for observations that first appeared after 25 million steps with more than or equal to 25 times.**
> > > > >
> > > > > Model | Mean | 68% Confidence Interval | 95% Confidence Interval | The number of observations
> > > > > |--|--:|--:|--:|--:|
> > > > > RND run1 | 2.14e-5 | (8.43e-6, 3.51e-5) | (-1.45e-5, 5.67e-5) | 308
> > > > > RND run2 | 2.35e-5 | (2.03e-5, 3.80e-5) | (-2.10e-5, 5.12e-5) | 354
> > > > > RND run3 | 2.89e-5 | (2.28e-5, 4.65e-5) |(-5.08e-5, 7.00e-5) | 114
> > > > > FarCuriosity run1 | -4.44e-5 | (-2.33e-5, 9.04e-6) | (-6.75e-4, 2.25e-5) | 312
> > > > > FarCuriosity run2 | -2.77e-4 | (-5.21e-5, 9.96e-6) | (-2.44e-3, 2.00e-5) | 238
> > > > > FarCuriosity run3 | -4.10e-3 |  (-8.43e-5, 1.51e-5) | (-7.01e-4, 1.31e-4) | 263
> > > > >
> > > > >
> > > > >
> > > > > We also collected intrinsic rewards for observations that appear in all runs (3 RND and 3 FarCuriosity) after one million frames and at least five times. We ran linear regression between the number of frames and intrinsic reward for each observation for each run to get linear coefficients as shown in Table R7.
> > > > >
> > > > > The coefficient for each model is different per observation. For example, intrinsic reward for observation 1 is reducing on average in each model. However, there are more positive coefficients in the other observations. For example, the intrinsic reward for observation 2 is generally increasing in RND while decreasing in FarCuriosity.
> > > > >
> > > > > In summary, although FarCuriosity does not completely solve catastrophic forgetting, it is much more robust than RND.
> > > > >
> > > > > **Table R7 Coefficient of linear regression for each observation that appears after 1 million frames at least five times in all runs.**
> > > > >
> > > > > Observation | RND | FarCuriosity
> > > > > |:---|---:|---:|
> > > > > Observation 1 | (-2.48e-4, -8.12e-5, -4.33e-5) | (-4.59e-5, -2.81e-5, -2.58e-5)
> > > > > Observation 2 | (3.34e-5, 3.81e-5, 4.13e-5) | (-1.39e-4, -6.33e-6, -3.64e-6)
> > > > > Observation 3 | (1.69e-5, 2.01e-5, 2.13e-5) | (-1.63e-4, 8.25e-6, 2.61e-5)
> > > > > Observation 4 | (2.17e-5, 4.69e-5, 1.57e-4) | (-6.36e-5, 2.79e-5, 1.97e-4)
> > > > >
> > > > > *Please let us know if you have any other concerns.*

---

### Author Response · Authors · 2022-11-14
**General Response**

We thank all the reviewers for their constructive comments. We revised the manuscript to reflect all comments, including an updated detailed description of our method and additional experiments. The changes are colored in blue. We will release the code if the paper is accepted. The main changes are as follows:

1. Clarify Section 3.1. FarMap, and add Figure 8 describing how to spatially transform the observation and explaining why the map size is increased.
2. Add map coverage and memory usage changes at each time step in the small / medium / large environments separately in Figure 5 (Figure 10 in the revised version).
3. Move Figure 11 (relative performance improvement) and corresponding analysis from the appendix to the main manuscript.
4. Add experimental results on harder Atari games; Gravitar, Montezuma’s Revenge, Pitfall, and Private Eye in Figures 6 and 11 (Figures 13 and 6 in the revised version).
5. Add a plot that shows the relationship between the number of fragmentations and performance (Figure 12 in the revised version).

---

### Decision · Program_Chairs · 2023-01-20

**Decision:**

Reject

**Justification For Why Not Higher Score:**

3 out of 4 reviewers said the paper should be rejected and were not willing to consider a different outcome. The reviewer who gave the paper a weak accept was very clear that they didn't want to go beyond that or even defend the paper, it felt more like a weak reject. Thus, unless I were to override the decision of 4 reviewers, which I was not comfortable doing by myself, I don't see how I could accept the paper.

**Justification For Why Not Lower Score:**

N/A.

**Metareview: Summary, Strengths And Weaknesses:**

This paper proposes the idea of fragmentation and recall and it demonstrates that it can be applied both in mapping problems and in reinforcement learning. The idea is to have small models that learn how to predict the agent’s next observation. When the model fails to do so, a fragmentation happens and a new model is created to focus on that region of the environment. At the end of the day the agent ends up with multiple local models that capture specific regions of the environment. At every step there’s also the possibility that the agent is actually seeing something that a different model captures really well, and that is the moment that the recall happens. The local models are connected as a graph that facilitates exploration across regions. It is said that such a split avoids catastrophic forgetting of having a single network learning a model of the whole world.

There was a lot of back and forth about this paper, and the paper has addressed several concerns raised by the authors, which the authors did appreciate. Regardless of the outcome I believe the reviewing process worked in making this paper better. Unfortunately, all authors unanimously agreed that a central claim tied to the motivation behind the proposed idea, which is to prevent catastrophic forgetting, was not properly backed up with experiments. The reviewers wanted to see some experiments demonstrating how the algorithm works, instead of jumping directly to justifying the method through performance. All in all, the final conclusion was that there was still a “lack of understanding about the basic functioning of the algorithm”.

I believe the paper is almost ready to be accepted. This paper will be very strong if the authors address these concerns about presentation/motivation, introducing simpler experiments to demonstrate how the algorithm works and to better justify claims around catastrophic forgetting. The reviewers were quite adamant that the paper is not ready for publication.

**Summary Of Ac-Reviewer Meeting:**

I did ask for a review in this particular case, because the paper had an average score of 5.25 and because no one seemed to feel strongly about it. I received a lot of push back from the reviewers though, they all agreed that the outcome of the meeting was predictable: the paper being rejected. Given that they expressed they were not anticipating a change in score, in any possible way, I didn't schedule the meeting.